# Regional variability of diatoms in ice cores from the Antarctic Peninsula and Ellsworth Land, Antarctica

Dieter R. Tetzner[1,2], Claire S. Allen[1], Elizabeth R. Thomas[1]

[1]British Antarctic Survey, Ice Dynamics and Paleoclimate, Cambridge, CB3 0ET, UK
[2]Department of Earth Sciences, University of Cambridge, Cambridge, CB2 3EQ, UK

*Correspondence to*: Dieter R. Tetzner (dietet95@bas.ac.uk)

**Abstract.** The presence of marine microfossils (diatoms) in glacier ice and ice cores has been documented from numerous sites in Antarctica, Greenland, as well as from sites in the Andes and the Altai mountains, and attributed to entrainment and transport by winds. However, their presence and diversity in snow and ice, especially in polar regions, is not well documented and still poorly understood. Here we present the first data to resolve the regional and temporal distribution of diatoms in ice cores, spanning a 20 year period across four sites in the Antarctic Peninsula and Ellsworth Land, Antarctica. We assess the regional variability in diatom composition and abundance at annual and sub-annual resolution across all four sites. These data corroborate the prevalence of contemporary marine diatoms in Antarctic Peninsula ice cores, reveal that the timing and amount of diatoms deposited vary between low and high elevation sites and support existing evidence that marine diatoms have the potential to yield a novel paleoenvironmental proxy for ice cores in Antarctica.

## 1 Introduction

Diatoms are unicellular algae with siliceous cell walls that inhabit aquatic environments throughout the world (Smol and Stoermer, 2010). Diatoms are abundant and diverse in lakes (Esposito et al., 2008; Verleyen et al., 2021), streams (Esposito et al., 2008; Noga et al., 2020), cryoconite holes (Stanish et al., 2013; Weisleitner et al., 2020), wet habitats (e.g. soils) (Van de Vijver and Beyens, 1998; Cavacini, 2001) and the oceans in general, especially in the Southern Ocean (SO) (Zielinski and Gersonde, 1997; Armand et al., 2005; Crosta et al., 2005; Alvain et al., 2008). Diatoms are particularly sensitive to oceanographic conditions and responsive to environmental changes. When diatoms die, they sink, promoting carbon export from the sea surface to deep waters. Moreover, the silicified nature of these cells allows their preservation in sediments and ice. These characteristics make them valuable as proxies for paleoenvironmental and palaeoceanographic reconstructions (Smol and Stoermer, 2010). Despite their aquatic habitats, several studies support they can be airborne (Lichti-Federovich, 1984; Gayley et al., 1989; Chalmers et al., 1996; McKay et al., 2008; Wang et al., 2008; Harper and Mckay, 2010; Spaulding et al., 2010; Hausmann et al., 2011; Budgeon et al., 2012; Papina et al., 2013; Fritz et al., 2015; Marks et al., 2019). Diatoms can be effectively lifted from the sea-surface microlayer into the atmosphere by wind-induced bubble-bursting and wave-breaking processes (Cipriano and Blanchard, 1981; Farmer et al., 1993). Once in the atmosphere, they can be transported by

winds over long distances (ranging from few kilometres to intercontinental scales) (Gayley et al., 1989; McKay et al., 2008; Harper and McKay, 2010). In Polar Regions, airborne diatoms can be deposited over ice sheets (Budgeon et al., 2012; Allen et al., 2020) and then buried under subsequent snowfall events to finally become part of the ice matrix.

Numerous studies have found wind-blown diatoms in air samples (Chalmers, 1996; McKay et al., 2008), fresh snow samples (Bleakley, 1996; Elster et al., 2007; McKay et al., 2008; Budgeon et al., 2012), exposed surfaces and in Antarctic ice cores
(Burckle et al., 1988; Kellogg and Kellogg, 1996; Delmonte et al., 2013; Barrett 2013; Delmonte et al., 2017; Allen et al., 2020; Tetzner et al., 2021a). Recently, Allen et al. (2020) presented preliminary evidence that diatoms recovered from an Ellsworth Land (EL) ice core may provide a novel proxy of past south westerly wind (SWW) strength over the SO. Despite the potential the diatom record has shown to reconstruct past wind strength, this proxy has only been evaluated on a single site, without accounting for any regional variability. Moreover, recent findings reported by Tetzner et al. (2021a) suggest the
diatom record preserved in ice cores from the Antarctic Peninsula (AP) exhibit intra-annual variations which could potentially dominate the annual signal, highlighting the need to study both annual and sub annual variability.

In this study, we present the diatom records from four shallow depth ice cores drilled between 2006 and 2020 in the AP and EL regions (Figure 1). Annual and sub-annual records of diatom abundance, concentration and assemblage composition are used to determine the regional and temporal variability of diatom content across the four sites. Ecological associations of the
diatoms present are used to identify likely source regions and potential transport pathways.

## 2 Regional Settings

The Antarctic continent and the ocean surrounding it present a region of contrasting environmental conditions. This section describes the main environmental features present in the AP-EL region and in the neighbouring Amundsen-Bellingshausen Seas (ABS).

**2.1 Oceanography**

The SO is a vast circumpolar region that encompasses the southern-most basins of the Atlantic, Indian and Pacific Oceans. Zonally, the SO can be sub-divided into two major circum-Antarctic regions delineated by the presence of oceanographic fronts and sea ice cover: The Northern Antarctic Zone (NAZ) and the Southern Antarctic Zone (SAZ) (Figure 1). The NAZ is characterized by year-round open waters, limited to the north by the Sub-Antarctic Front (SAF) and to the south by the
maximum extent of seasonal sea ice cover. Within the NAZ is the Permanently Open Ocean Zone (POOZ), a region presenting Antarctic surface waters year-round, delimited by the Antarctic Polar Front (APF) to the north and by the Seasonal Sea Ice Edge (SSIE) to the south. The SAZ is characterized by the presence and variability of seasonal sea ice cover, delineated by the austral winter sea ice maximum to the north and by the Antarctic coast to the south. Within the SAZ is the Seasonal Sea Ice Zone (SSIZ), the oceanographic region subject to the sea ice annual cycle, delimited by the SSIE to
the north and by the Perennial Sea Ice Edge (PSIE) to the south.

The SO is one of the most productive water masses on Earth. The spatial and temporal patterns of productivity are highly variable across the region but can be generalised into two regions that broadly coincide with the NAZ and SAZ (Arrigo et al., 2008; Soppa et al., 2016). Primary production in the NAZ is characterized by a moderate seasonal cycle, highest (lowest) during austral spring (winter) at 300–400 (50-70) mg C $m^{-2}$ $d^{-1}$, and fairly stable annual production with interannual

variability of 3.5 % (Arrigo et al., 2008). Conversely, primary productivity in the SAZ presents a strong seasonal cycle characterised by short-lived, intense blooms during austral spring/summer (>1600 mg C $m^{-2}$ $d^{-1}$) (Rousseaux and Gregg, 2013; Arrigo et al., 2008; Soppa et al., 2016). These intense blooms are triggered by increasing light availability and melt-induced stratification during the austral spring and/or summer months (Soppa et al., 2016). The opposite happens during the austral winter, when productivity in the SAZ is at its lowest due to light limitation and the large area covered by sea ice

(Arrigo et al., 1998). Interannual variability in SAZ primary productivity (>19%) is considerably higher than the interannual variability observed in the NAZ (3.5 %), mainly driven by changes in the distribution and timing of sea ice melt (Arrigo et al., 2008; Smith and Comiso, 2008).

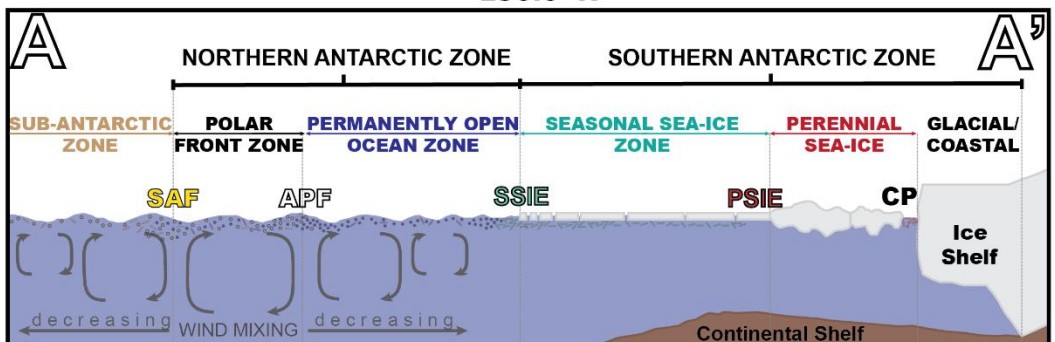

**Figure 1.** Map showing the ice core sites and oceanographic features considered in this study. The yellow circles show the locations of the four ice core sites. The red circle show the location of the Ferrigno ice core (Allen et al., 2020). The pale green line shows the median September sea ice extension between 1980-2010 CE and represents the Seasonal Sea Ice Edge (SSIE). The red line shows the median February sea ice extension between 1980-2010 CE, and represents the Perennial Sea Ice Edge (PSIE). A-A' presents a schematic cross-section including the standard oceanographic features across the Amundsen-Bellingshausen Sea (Schematic cross-section adapted from Allen et al. (2011)). Gray box delineate the region presented in Figure 6. SAF= Sub-Antarctic Front. APF= Antarctic Polar Front. SSIZ= Seasonal Sea Ice Zone. PSIZ= Perennial Sea Ice Zone. CP= Coastal Polynya. AS= Amundsen Sea. BS= Bellingshausen Sea. EL= Ellsworth Land. ROIC= Rothschild ice core site. JUR= Jurassic ice core site. SKBL= Sky-Blu ice core site. SHIC= Sherman Island ice core site.

## 2.2 Climate

Regional atmospheric circulation in the AP-EL and the neighbouring ABS is dominated by the Amundsen Sea Low (ASL) with SWW advecting warm and moist air from the southern Pacific Ocean towards the AP (Turner et al., 2013; Orr et al., 2004). When approaching the Antarctic Peninsula, airmasses are diverted south, producing a northerly wind flow. Once airmasses reach the Antarctic ice sheet, they are blocked and deflected to the east where the flow is enhanced by katabatic winds flowing downslope from the ice sheet interior. This easterly flow is known as the Antarctic coastal easterlies (Hazel and Stewart, 2019). Winds over the SO, including the ABS, present a clear seasonality with stronger winds during the austral winter and weaker winds during the austral summer (Yu et al., 2020, Thomas and Bracegirdle, 2015; van Wessem et al., 2015). During the satellite-era (1979-present), surface winds have strengthened across the AP-EL region (van Wessem et al., 2015), corresponding to the broader observation that the SWW have experienced the strongest positive trend worldwide (Young and Ribal, 2019).

Air temperatures present a regional gradient with temperatures decreasing with increasing latitude and elevation. This regional gradient leaves coastal regions particularly sensitive to positive degree days during the austral summer with surface melting mainly restricted to coastal areas below 400 m a.s.l. (van Wessem et al., 2015; van Wessem et al., 2016). Direct temperature measurements and regional atmospheric climate models show temperatures have followed a positive trend in the southern AP region during the second half of the twentieth century (Gonzales and Fortuny, 2018) and a slightly negative trend in the EL region (van Wessem et el., 2015).

Precipitation in the AP-EL regions are relatively constant throughout the year, exhibiting slightly lower values during the austral summer (Thomas and Bracegirdle, 2015; van Wessemen, 2016). A regional gradient of decreasing precipitation with increasing elevation is identified across the region (van Wessem et al., 2016). Precipitation patterns in the AP-EL region are not considerably influenced by the occurrence of extreme precipitation events (Turner et al., 2019). Ice core records have demonstrated the AP-EL region has experienced a long-term positive trend in snow accumulation over the twentieth century (Thomas and Tetzner, 2018), with some sites even doubling their snow accumulation (Thomas et al., 2008).

## 2.3 Sea ice

The Antarctic sea ice cover exhibits a regular seasonal cycle presenting its maximum (minimum) surface extension during September (February) (Parkinson, 2019). During the satellite-era (1979-present), the total Antarctic sea ice area has increased between 1.0 and 1.5 % (Parkinson and Cavalieri, 2012; Parkinson, 2019). However, this trend masks some abrupt interannual variations and diverse regional processes. In the ABS sectors, there has been a sustained, significant decrease in the total area covered by sea ice (−2.5 % per decade) (Parkinson, 2019). This decline has led to an earlier retreat and a delayed formation of the sea ice cover, which together have resulted in a 3-month extension to the austral summer ice-free season in the ABS (Stammerjohn et al., 2012).

## 2.4 Terrestrial environment

Ice-free areas represent less than 3% of the AP-EL surface and mostly comprise of nunataks (Siegert et al., 2019). Over the last few decades, accelerated glacier retreat throughout the AP (Cook et al., 2005; Seehaus et al., 2018) has favoured the expansion of ice-free areas. This has promoted the formation of lakes, meltwater streams and soil development in the newly ice-free terrain, all subject to seasonal and/or diurnal freeze and thaw (Ruiz-Fernández et al., 2019). Despite the widespread glacier retreat across the AP, the recent formation of these landforms is mostly constrained to the northern Antarctic Peninsula (Ruiz-Fernández et al., 2019). Additionally, recent increases in air temperatures and strengthened poleward flow on the western Antarctic Peninsula have promoted the formation of seasonal supraglacial water bodies on top of glaciers and ice shelves (Dirscherl et al., 2021).

## 3 Methods

### 3.1 Ice core records and age scales

Four ice cores from the southern AP and EL were included in this study (Figure 1) (Table 1). The Sherman Island ice core (SHIC, 21.3 m) from the West Antarctic ice sheet coast, and the Sky-Blu ice core (SKBL, 21.8 m) from the vicinity of Sky-Blu Field Station, southern AP, were both drilled using a Kovacs hand-auger during the austral summer 2019/2020. The Rothschild ice core (ROIC, 11.1 m) from Rothschild Island, southern AP, was drilled using a Kovacs hand-auger during the austral summer 2005/2006. The Jurassic core (JUR, 140 m) from an inland site of the English Coast, southern AP, was drilled using the BAS electromechanical drill during the austral summer 2012/2013. For SHIC, SKBL and JUR, an ice core chronology was established based on their hydrogen peroxide ($H_2O_2$) annual cycle that is assumed to peak during the austral summer solstice and to exhibit its minimum during the austral winter (Frey et al., 2006; Thomas et al., 2008). Ice core chronologies were resolved using the annual cycle of the non-sea salt component of major ions, such as non-sea salt sulphates (nss$SO_4^{2-}$) (Piel et al., 2006), that is assumed to peak between November and January in this region (Pasteris et al., 2014; Thoen et al., 2018). This non-sea salt stratigraphy was further corroborated by the presence of volcanic tephra in the

2001 CE ice core layer (Tetzner et al., 2021b). The top 15 m of SKBL included in this work and the full SHIC core were dated back to 1999 CE, with an estimated dating error for the 1999-2020 CE interval of ±3 months for each year and with no accumulated error. For ROIC, the ice core was dated using the annual cycles of major ion concentrations, resulting in an age scale from 2002-2006 CE. All annual values are reported as the austral winter-to-winter phase.

**Table 1. Summary of each ice core geographical location and main features of the datasets analysed in this study. SIE= Sea Ice Edge(*) - The distance from SIE reported corresponds to the median for years covering the data interval (See section 3.3). September SIE values used for calculations were obtained as distance between the ice core site and the closest point in the northern limit of 15% sea ice cover. February SIE values used for calculations were obtained as the distance between the ice core site and the closest sea ice free region.**

| Core name | Long | Lat | Elevation (m a.s.l) | ANNUAL RECORD | | SUB-ANNUAL RECORD | | Total depth used (m) | Distance from SIE (km)* | |
|---|---|---|---|---|---|---|---|---|---|---|
| | | | | Years (CE) | # samples | Years (CE) | # samples | | Sept | Feb |
| JUR | -73.06 | -74.33 | 1139 | 1992-2012 | 20 | 2002-2006 | 16 | 36.9 | 1045 | 140 |
| SKBL | -71.59 | -74.85 | 1419 | 1999-2019 | 20 | 2002-2006 | 32 | 15.0 | 1148 | 200 |
| SHIC | -99.63 | -72.67 | 474 | 1999-2019 | 20 | 2002-2006 | 16 | 21.3 | 753 | 130 |
| ROIC | -72.6 | -69.6 | 438 | - | - | 2002-2006 | 36 | 11.1 | 598 | 70 |

## 3.2 Sample preparation and analyses

All ice cores included in this study were cut, using a band-saw with a steel blade, to obtain up to four ice core strips. The first strip (2 x 4 cm) was sub-sampled at 5 cm resolution and processed for ion chromatographic analyses of major ions and Methanesulphonic Acid (MSA) using a reagent-free Dionex ICS-2500 anion and IC 2000 cation system in a class-100 cleanroom. The MSA is the oxidized product of DMS, an organic sulphur compound from marine biogenic emissions. Measurements of MSA were used to estimate the temporality of the ROIC ice core. The MSA record was used as it has demonstrated to present a clear seasonal cycle in Antarctic ice cores, with a sharp austral summer maximum and a broad winter trough (Abram et al., 2013).

A second ice core strip (3.3 x 3.3 cm) was cut from SHIC, SKBL and JUR and then melted using a Continuous Flow Analysis (CFA) system (Rothlisberger et al., 2000; Grieman et al., 2021) in the ice chemistry lab at the British Antarctic Survey, UK to analyse the $H_2O_2$ concentration using enzymatic fluorometry examined by a FIAlab photomultiplier-FL detector through a 3 mm Suprasil flow cell.

A third ice core strip was used for the diatom analyses. For SHIC, SKBL and JUR, this third strip was cut at annual resolution and an additional fourth ice core strip was cut at sub-annual resolution. Sub-annual samples for SHIC and JUR were cut based on the position of the $H_2O_2$ austral summer maxima and austral winter minima. Each annual interval between the $H_2O_2$ maxima and minima in SHIC and JUR, was split to obtain four sub-annual samples per year. Sub-annual samples from SKBL were cut at 10 cm resolution (~7-8 samples per year). Sub-annual samples from ROIC were cut at 30 cm

resolution (~9 samples per year). Seasons are reported as austral summer (December to February, DJF), austral autumn (March to May, MAM), austral winter (June to August, JJA) and austral spring (September to November, SON).

All diatom samples were processed and analysed following the method and recommendations presented in Tetzner et al. (2021a). Observations regarding diatom preservation were based on the characteristics of frustule dissolution and degradation described by Warnock and Scherer (2015). Diatom frustules and fragments with a long axis less than 5 μm were excluded from the diatom counting and identification.

After processing, diatom counts per sample (n) were transformed to diatom abundance (n t$^{-1}$), where "t" represents the
temporal resolution of each sample. To compare the magnitude of the diatom abundance in different ice core sites, diatom concentrations (n L$^{-1}$) were calculated by normalizing the diatom counts per sample (n) with the meltwater volume (L) filtered. All correlations reported in this work were calculated after detrending each dataset and were calculated using the Pearson's linear correlation (R). All timeseries linear correlations were calculated over a 20-year period (1992-2012 CE for JUR and 1999–2019 CE for SHIC and SKBL).

Diatom identification and ecological associations were based on Armand et al. (2005), Halse and Syvertsen (1996), Hasle and Syvertsen (1997), Cefarelli et al. (2010), Zielinkski and Gersonde (1997) and references therein. Diatoms were identified to species level where possible. Diatoms that were not possible to identify to species level due to insufficient image resolution which made recognition of diagnostic features difficult, were combined in genera/morphological groups. Among the diatoms identified as *Fragilariopsis cylindrus* it is noted that they may also include *F. nana* (distinguished from *F.*
*cylindrus* by size as per Cefarelli et al., 2010). Diatoms that were not possible to classify (ie. Valves partly obscured by insoluble particles lying on top, fragments with undiagnostic features & poorly ornamented or indistinct fragments) were omitted from ecological associations and assemblage composition but were included in the total diatom counts (n). Diatoms that were not possible to classify were broadly grouped as pennates or centrics based on the symmetry of the valve from which they likely originated (Hasle and Tomas, 1997). These diatoms were grouped in this way to identify if they were
preferentially fragmented based on their crude morphology. The assemblage composition was determined for each site from the identified species and groups with abundances higher than 2.0 % of the whole assemblage and present in at least two samples of the 20-year record. Ecological associations were determined for the most abundant species/groups of each core. For the three 20-year diatom records (SHIC, JUR and SKBL), the assemblage composition was analysed over the whole period and for the two decadal subsets. The decadal subsets were produced to study temporal changes in diatom relative
abundance and concentration over shorter timescales in order to assess the consistency of the assemblage in response to recent environmental changes in the region. The assemblage composition at ROIC was only analysed over the 4-year period (2002-2006 CE). Sub-annual comparisons of the diatom relative abundance and diatom concentration were made over the common overlapping interval for all four sites (January 2002- January 2006 CE) (Table 1) to analyse the regional intra-annual variations in the diatom record. A Sea Ice Diatom Index (SIDI) was calculated for each sub-annual sample as the sum
of the diatom concentrations of the two characteristic sea ice diatoms in the SO: *F. cylindrus* and *F.curta* (Lizotte, 2001;

Armand et al., 2005). The SIDI from each ice core site was analysed over the overlapping period to study the relation between the total diatom concentration and the sea ice diatom concentration.

## 3.3 Sea ice extension data

Sea ice extent data were obtained from the satellite derived Sea Ice Index, Version 3 dataset (Fetterer et al., 2017) from the National Snow and Ice Data Centre (NSIDC). The Sea Ice Index provides monthly data on sea ice concentrations available at 25 km resolution from 1979 onward. September sea ice limits (defined as the median northerly extent of 15 % sea ice cover) were considered as the annual sea ice maximum, while February sea ice limits (defined as the median northerly extent of 15 % sea ice cover) were considered as the annual sea ice minimum (Thomas et al., 2019).

## 4 Results

A total of 4437 diatom valves and fragments were found among all samples. Of them, 2811 were found in annual samples, while 1626 were found in sub-annual samples. Diatoms were well preserved, with no evidence of dissolution in their structure, preserving delicate ornamentation and occurring as colonies of up to five cells. No clear trend was identified in the proportion of fragments relative to diatom frustules down-core. Similarly, no clear preference or tendency was identified in the proportion of pennate fragments relative to centric fragments down-core. The main features and basic statistics of the diatom record for each ice core site are presented in Table 2.

A total of 25 diatom species and generic/taxa groupings were identified among all ice core sites. Of them, ten occurred at >2 % relative abundance in at least two samples of an ice core. Of these ten main taxa, six were present in more than one site, four were present in samples across all four ice core sites and four occurred exclusively at one site. Table 3 presents the relative abundance data for the ten main taxa in each ice core. Table 4 presents the basic statistics of the annual diatom abundance and concentration for each ice core site.

**Table 2. Main features and basic statistics of the annual and sub-annual diatom records for each ice core. *Calculations excluding austral spring 2002 CE. +Annual ROIC values were calculated combining sub-annual samples based on ROIC chronology (See section 2.1).**

| Core name | Number of samples | Total Diatom counts (n) | Total diatoms classifiable | Total volume filtered (L) | Mean volume filtered ($\bar{x}$ ± s.d.) |
|---|---|---|---|---|---|
| **Annual record** | | | | | |
| SHIC | 20 | 1087 | 822 | 5.76 | 0.288 ± 0.069 |
| JUR | 20 | 1140 | 544 | 6.89 | 0.345 ± 0.079 |
| SKBL | 20 | 584 | 475 | 5.05 | 0.252 ± 0.086 |
| ROIC[+] | 4 | 665 | 435 | 3.78 | 0.943 ± 0.186 |

| Sub-annual record (2002-2006 CE) | | | | | |
|---|---|---|---|---|---|
| SHIC | 16 | 594 | 499 | 1.69 | 0.106 ± 0.03 |
| JUR* | 16 | 226 | 91 | 1.33 | 0.089 ± 0.014 |
| SKBL | 32 | 164 | 42 | 2.92 | 0.091 ± 0.007 |
| ROIC | 36 | 665 | 435 | 3.78 | 0.105 ± 0.014 |

**Table 3. Relative abundance (%) and frequency (# of samples) of main diatom taxa in annual and sub-annual diatom records for each ice core. (\*) specimens of *Cyclotella* sensu lato (including *Lindavia*, *Discostella*, *Tertiarius* and *Pantocsekiella* and other morphologically similar types). n represents the total number of samples in each sub-annual record. (s)= sea ice affiliated diatom. (o-SSIE)= open ocean – Seasonal Sea Ice Edge affiliated diatom. (o-POOZ)= open ocean – Permanently Open Ocean Zone affiliated diatom species/group.**

| | SHIC | JUR | SKBL | ROIC |
|---|---|---|---|---|
| **Annual record** | **(1999-2019 CE)** | **(1992-2012 CE)** | **(1999-2019 CE)** | |
| *Fragilariopsis cylindrus* (s) | 63.7 % (20) | 18.2 % (14) | 21.3 % (15) | - |
| *Shionodiscus gracilis* (o-SSIE) | 18.5 % (18) | 17.6 % (17) | 10.9 % (10) | - |
| *Fragilariopsis curta* (s) | 4.1 % (10) | - | - | - |
| *Fragilariopsis pseudonana* (o-POOZ) | 3.6 % (11) | 9.2 % (11) | 15.3 % (10) | - |
| *Cyclotella* group* | 6.8 % (19) | 29.1 % (19) | 37.2 % (20) | - |
| *Thalassiothrix* group (o-POOZ) | - | - | - | - |
| *Navicula* group | - | 7.4 % (12) | - | - |
| *Nitzschia* group | - | - | 6 % (12) | - |
| *Pseudonitzschia* spp. (o-POOZ) | 3.3 % (10) | 6.5 % (7) | 9.3 % (8) | - |
| *Achnanthes* group | - | 11.9 % (10) | - | - |
| **Sub-annual record (2002-2006 CE)** | **(n=16)** | **(n=16)** | **(n=32)** | **(n=36)** |
| *Fragilariopsis cylindrus* (s) | 73.8 % (14) | 34 % (11) | 23.1 % (4) | 83 % (21) |
| *Shionodiscus gracilis* (o-SSIE) | 7.1 % (7) | 8.8 % (4) | 7.7 % (2) | 6.5 % (11) |
| *Fragilariopsis curta* (s) | 8.1 % (6) | - | - | 4 % (10) |
| *Fragilariopsis pseudonana* (o-POOZ) | 4.2 % (4) | 14.3 % (4) | 3.8 % (1) | 2.5 % (6) |
| *Cyclotella* group* | 3.1 % (6) | 21.1 % (13) | 34.6 % (5) | 2 % (6) |

| | | | | |
|---|---|---|---|---|
| *Thalassiothrix* group (o-POOZ) | - | - | - | 2 % (6) |
| *Navicula* group | - | 6.8 % (5) | - | - |
| *Nitzschia* group | - | - | 7.7 % (2) | - |
| *Pseudonitzschia* spp. (o-POOZ) | 3.7 % (6) | 15 % (7) | 23.1 % (5) | - |
| *Achnanthes* group | - | - | - | - |

**Table 4. Basic statistics for the diatom abundance and diatom concentration records from each ice core. *Calculations excluding austral spring 2002 CE. +Annual ROIC values were calculated combining sub-annual samples based on ROIC chronology (See section 2.1).**

| Core Name | Time Interval (CE) | Diatom abundance (n yr$^{-1}$) | | | Diatom concentration (n L$^{-1}$) | | |
|---|---|---|---|---|---|---|---|
| | | Mean (± s.d.) | Max | Min | Mean (± s.d.) | Max | Min |
| **Annual record** | | | | | | | |
| SHIC | 1999-2019 | 54.4 (33.7) | 110 | 10 | 188.9 (110.0) | 376.3 | 41.8 |
| JUR | 1992-2012 | 57.0 (34.0) | 166 | 20 | 180.7 (143.7) | 723.0 | 52.5 |
| SKBL | 1999-2019 | 29.2 (14.9) | 77 | 15 | 130.4 (72.1) | 304.4 | 51.8 |
| ROIC$^+$ | 2002-2006 | 166.3 (42.1) | 226 | 129 | 181.4 (52.0) | 230.1 | 111.2 |
| **Sub-annual record** | | | | | | | |
| SHIC | 2002-2006 | 37.1 (50.1) | 170 | 4 | 431.1 (671.2) | 2538.1 | 34.8 |
| JUR* | 2002-2006 | 15.1 (5.8) | 25 | 7 | 168.3 (58.1) | 303.4 | 102.1 |
| SKBL | 2002-2006 | 5.1 (2.9) | 13 | 1 | 56.3 (31.4) | 148.7 | 10.3 |
| ROIC | 2002-2006 | 18.5 (19.1) | 89 | 0 | 178.2 (177.3) | 831.4 | 0 |

**4.1 Jurassic ice core (JUR)**

**4.1.1 Diatom record**

A total of 1140 diatom valves and fragments were identified in the JUR annual record (Table 2). The mean annual diatom abundance was 57 ± 34 n yr$^{-1}$, with annual diatom abundance values ranging from 20 n yr$^{-1}$ (2004-2005 CE) to 166 n yr$^{-1}$ (2009-2010 CE) (Figure 2b) (Table 4). Diatom abundance is not correlated with the volume of meltwater filtered per sample

(R=-0.19, p>0.05) or with the annual snow accumulation estimated from the ice core (R=-0.32, p>0.05). Mean annual diatom concentration was 180.7 ± 143.7 n L$^{-1}$, with annual diatom concentration values ranging from 52.5 n L$^{-1}$ (2004-2005

CE) to 723 n L$^{-1}$ (2009-2010 CE) (Figure 2b) (Table 4). The diatom concentration record exhibited a positive trend of 8.25 n L$^{-1}$ yr$^{-1}$ (p=0.14) over the 1992-2012 CE period with mean diatom concentrations 41.39% higher between 2002-2012 CE (compared with 1992-2002) (Figure 2b).

The sub-annual diatom record from JUR exhibited uniform diatom concentration values and a single, major, increase during the austral spring 2002 CE (1383.3 n L$^{-1}$). Since this sample increase was more than five times higher than the mean diatom concentration over the whole 2002-2006 CE period (244.3 ± 308.9 n L$^{-1}$) and not evident in the 20 year annual record of JUR, we consider this single major increase anomalous and exclude it from the sub-annual calculations. Excluding the increase identified during the austral spring of 2002 CE, the mean diatom concentration was 168.3 ± 58.1 n L$^{-1}$, showing

moderate variations for 2003-2005 CE and no clear seasonality (Figure 2c).

    Of the 1140 diatoms counted in the annual diatom record of the JUR ice core, 544 were identified to genus level or higher. Seven species/taxa groupings occur at >2% in at least two samples with highest relative abundances of *Shionodiscus gracilis* (17.6%), *Fragilariopsis cylindrus* (18.2%), and the *Cyclotella* group (29.1%) (**Table 3**). Decadal subsets show minor discrepancies, only exhibiting a 7% increase in the abundance of *S. gracilis* and a 6% decrease in the abundance of

250 *Achnanthes* group in the most recent decade (2002-2012 CE) (**Figure 2a**). Sub-annual samples showed no clear seasonality in the distribution of the SIDI over the 2002-2006 CE period (**Figure 2c**).

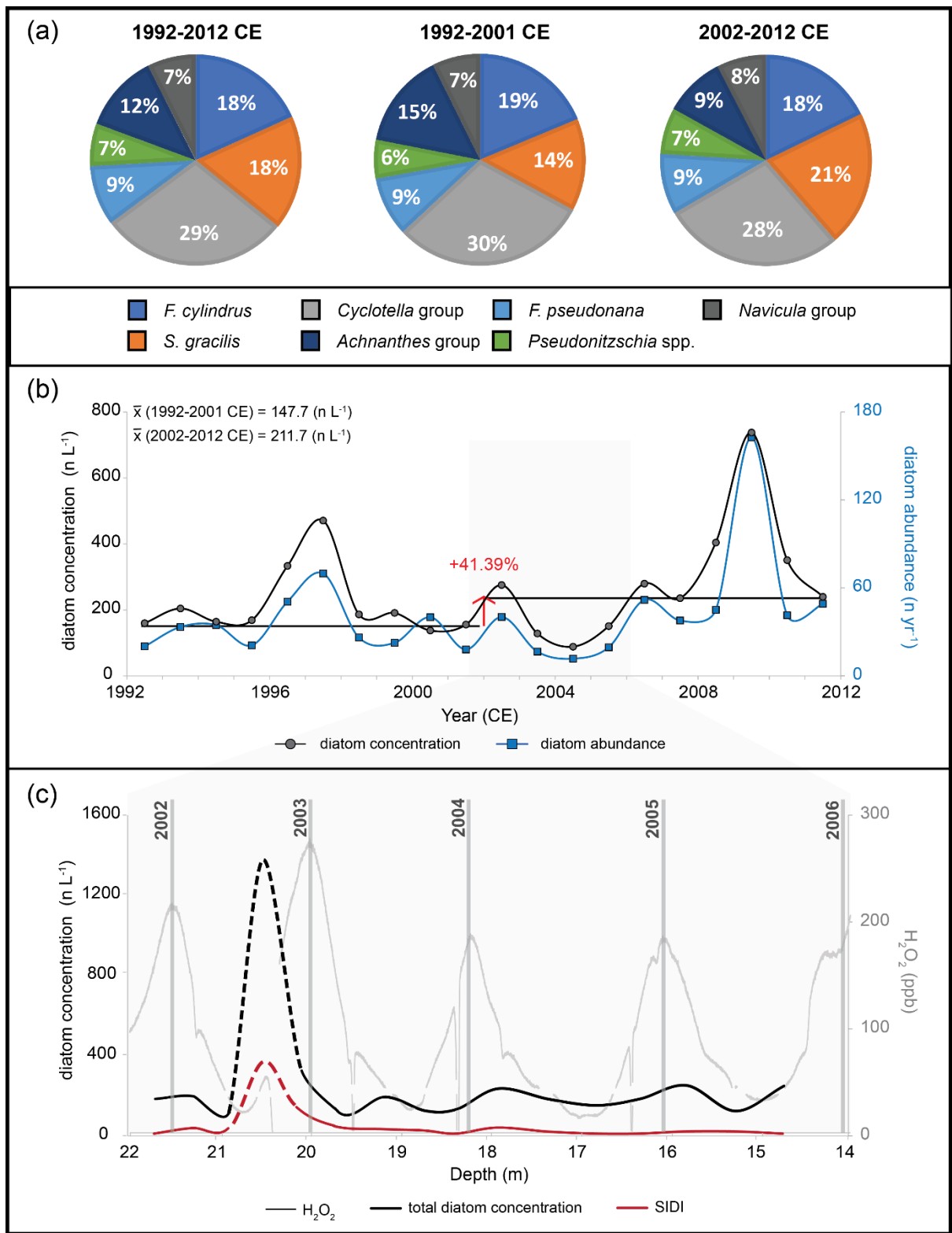

**Figure 2. Diatom record from the JUR ice core. a) Main diatom assemblage composition of the JUR ice core during 1992-2012 CE and during decadal subsets. Percentages reported in this figure were normalized to the main species identified. b) diatom concentration and diatom abundance timeseries. The red arrow represents the percentual variation in the diatom concentration decadal mean. c) Total diatom concentration and SIDI variations down-core for the 2002-2006 CE horizon. Vertical grey lines indicate temporal horizons based on the austral summer maxima in $H_2O_2$.**

### 4.2 Sky-Blu ice core (SKBL)

### 4.2.1 Diatom record

A total of 584 diatom valves and fragments were identified in SKBL (Table 2). A mean annual diatom abundance of $29.2 \pm 14.9$ n yr$^{-1}$ was obtained, with values ranging from 15 n yr$^{-1}$ (2009-2010 CE) to 77 n yr$^{-1}$ (2004-2005 CE) (Figure 3b) (Table 4). No correlation is observed between the diatom abundance and either the volume of meltwater filtered per sample (R=-0.01, p>0.05) or the annual snow accumulation estimated from the ice core (R=-0.03, p>0.05). Mean annual diatom concentration was $115.6 \pm 72.1$ n L$^{-1}$, with values ranging from 51.8 n L$^{-1}$ (2012-2013 CE) to 304.4 n L$^{-1}$ (2009-2010 CE) (Figure 3b) (Table 4). The annual diatom concentration record presented a positive trend of 2.56 n L$^{-1}$ yr$^{-1}$ (p=0.37) over the 1999-2019 CE period with an increase of 25.56% from a mean concentration of 115.6 n L$^{-1}$ for 1999-2008 CE, to 145.1 n L$^{-1}$ for 2009-2019 CE (Figure 3b). SKBL sub-annual diatom concentrations (mean=$56.3 \pm 34.4$ n L$^{-1}$) presented modest variations throughout the year with slightly higher concentrations consistently during late austral summer/early autumn (Figure 3c).

Among the 584 diatoms counted in SKBL, 475 were classified to genus level or higher. Six species/taxa groupings were present at >2% in at least two samples (Table 3). with highest relative abundances of *S. gracilis* (10.9%), *F. pseudonana* (15.3%), *F. cylindrus* (21.3%) and the *Cyclotella* group (37.2%). The most recent decade (2009-2019 CE) exhibited a decrease of 10% and 6% in the relative abundance of *Pseudonitzschia* spp. and the *Cyclotella* group, respectively and a 6% increase of both *S. gracilis* and *F. cylindrus* (Figure 3a). Sub-annual samples revealed no clear seasonality in the distribution of SIDI nor in-phase variability between SIDI and the total diatom counts over the 2002-2006 CE period (Figure 3c).

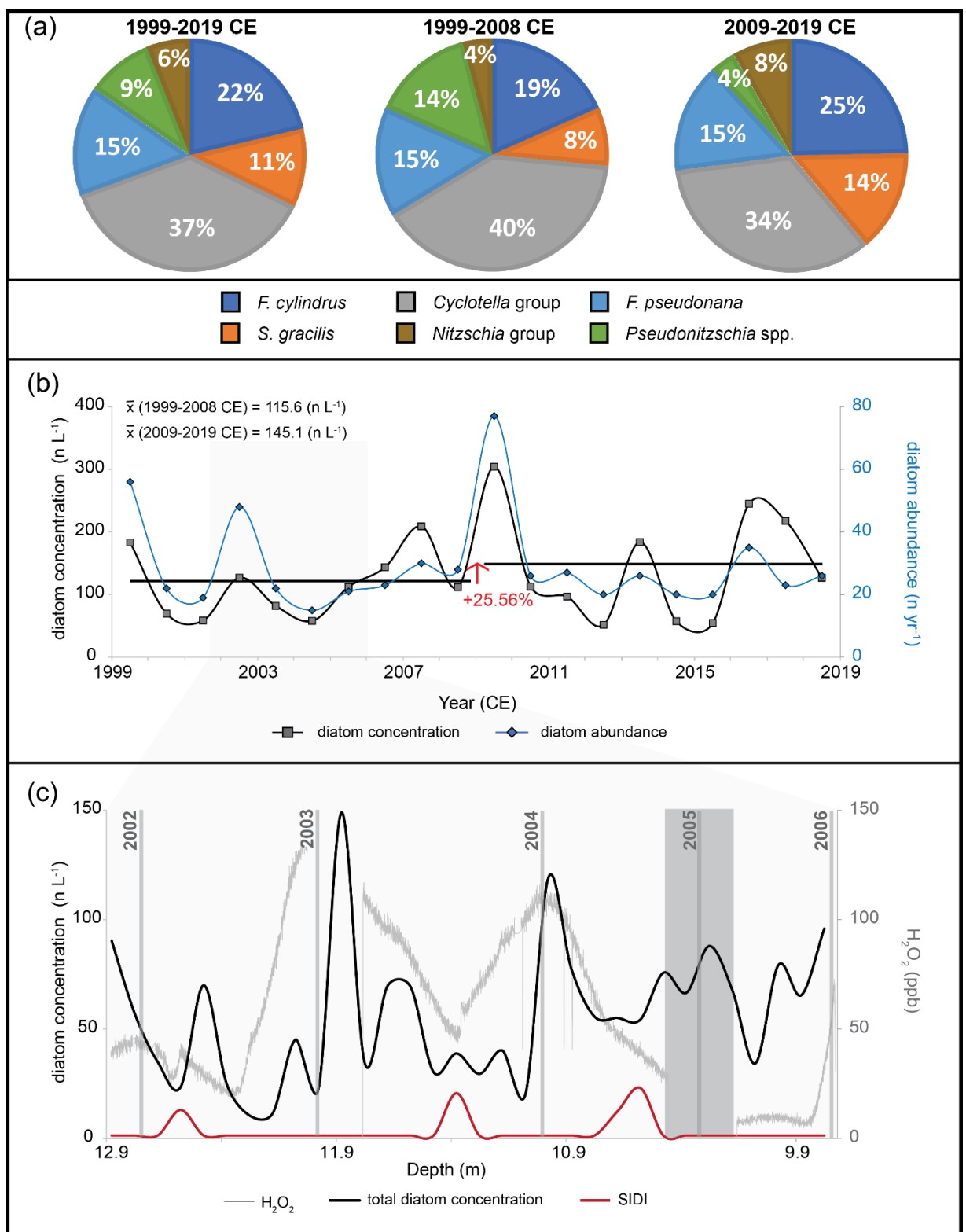

**Figure 3. Diatom record from the SKBL ice core. a) Main diatom assemblage composition of the SKBL ice core during 1999-2019 CE and during decadal subsets. Percentages reported in this figure were normalized to the main species identified. b) diatom concentration and diatom abundance timeseries. The red arrow represents the percentual variation in the diatom concentration decadal mean. c) Total diatom concentration and SIDI variations down-core for the 2002-2006 CE horizon. Vertical grey lines indicate temporal horizons based on the austral summer maxima in $H_2O_2$. Vertical grey band indicates a gap in the $H_2O_2$ dataset.**

## 4.3 Sherman Island ice core (SHIC)

### 4.3.1 Diatom record

The annual diatom record from SHIC comprised 1087 diatom valves and fragments (Table 2). The annual diatom abundance presented a mean value of $54.4 \pm 33.7$ n yr$^{-1}$, with annual values ranging from 10 n yr$^{-1}$ (2000-2001 CE) to 110 n yr$^{-1}$ (2014-2015 CE) (Figure 4b) (Table 4). A weak but not statistically significant correlation is observed between diatom abundance and meltwater filtered per sample (R=0.28, p>0.05), while no correlation is observed with the annual snow accumulation estimated from the ice core (R=-0.09, p>0.05). Mean annual diatom concentration was $188.9 \pm 110$ n L$^{-1}$, with values ranging from 41.8 n L$^{-1}$ (2000-2001 CE) to 376.3 n L$^{-1}$ (2010-2011 CE) (Figure 4b) (Table 4). The annual diatom concentration record presented a positive trend of 6.23 n L$^{-1}$ yr$^{-1}$ (p=0.15) over the 1999-2019 CE period. Decadal subset analyses showed an increase in the mean annual diatom concentration of 63.76% from 142.5 n L$^{-1}$ (1999-2008 CE) to 233.3 n L$^{-1}$ (2009-2019 CE) (Figure 4b). The sub-annual diatom concentration record from SHIC exhibited a clear seasonal pattern characterized by higher values (>300 n L$^{-1}$) during austral summer/early autumn, and lower values (<100 n L$^{-1}$) during austral winter/early spring (Figure 4c).

Of the 1087 diatoms counted in SHIC, 822 were identified to genus level or higher. Six species/taxa groupings occurred >2% in at least two samples of SHIC, with *F. cylindrus* (63.7%) and *S. gracilis* (18.5%) accounting for more than 82% of the diatoms identified at SHIC (Table 3). Decadal subsets from the main diatom assemblage show minor variations, the largest being a 5% decrease (increase) in the relative abundance of *F. curta* (*F. cylindrus*) in the most recent decade (2009-2019 CE) (Figure 4a). Sub-annual samples from SHIC exhibited seasonality in the distribution of the SIDI, in-phase with the seasonality described by the total diatom counts over the 2002-2006 CE period (Figure 4c).

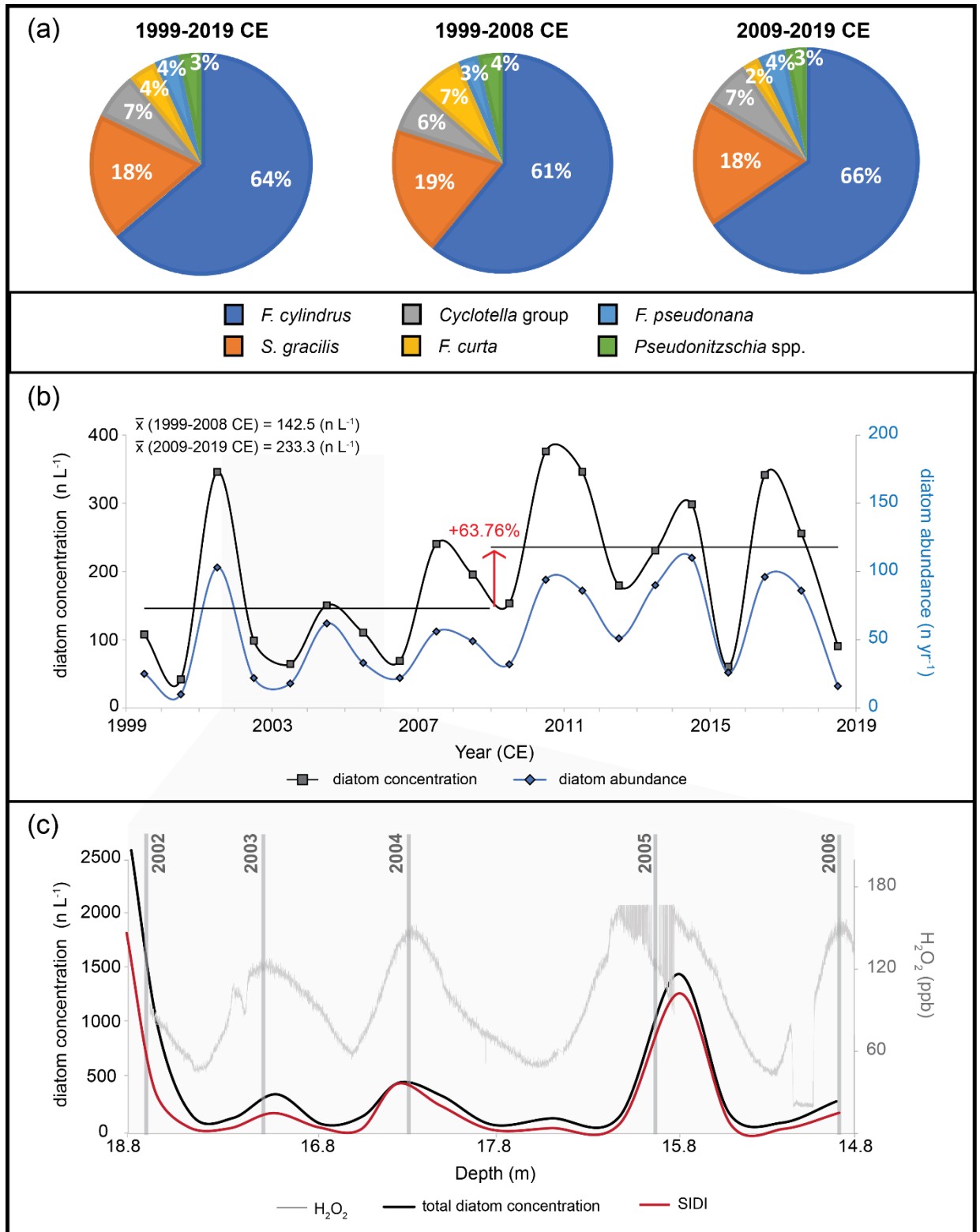

**Figure 4. Diatom record from the SHIC ice core. a) Main diatom assemblage composition of the SHIC ice core during 1999-2019 CE and during decadal subsets. Percentages reported in this figure were normalized to the main species identified. b) diatom concentration and diatom abundance timeseries. The red arrow represents the percentual variation in the diatom concentration decadal mean. c) Total diatom concentration and SIDI variations down-core for the 2002-2006 CE horizon. Vertical grey lines indicate temporal horizons based on the austral summer maxima in $H_2O_2$.**

## 4.4 Rothschild ice core (ROIC)

### 4.4.1 Diatom record

The diatom record from ROIC comprised 665 diatom valves and fragments (Table 2). The annual diatom abundance presented a mean value of $166.3 \pm 42.1$ n yr$^{-1}$, with values ranging from 129 n yr$^{-1}$ (2002 CE) to 226 n yr$^{-1}$ (2003 CE) (Table 4). Diatom abundance is not correlated with volume of meltwater filtered per sample (R=0.10, p>0.05). The annual diatom concentration presented a mean value of $176.1 \pm 52$ n L$^{-1}$, with values ranging from 111.2 n L$^{-1}$ (2002 CE) to 230.1 n L$^{-1}$ (2003 CE) (Table 4). The sub-annual diatom concentration from ROIC revealed a strong seasonality, with higher values (>250 n L$^{-1}$) during austral summer/early autumn, and lower values (<70 n L$^{-1}$) during austral winter/early spring (Figure 5b).

Among the 665 diatoms counted in ROIC, 435 were identified to genus level or higher. The assemblage included six diatom species/taxa, dominated by *F. cylindrus* (83%) with minor presence of *S. gracilis* (6.5%) and *F. curta* (4%) (Figure 5a) (Table 3). Sub-annual samples from ROIC presented a clear seasonality in the distribution of SIDI, in-phase with the seasonal pattern identified in the total diatom counts over the 2002-2006 CE period (Figure 5b).

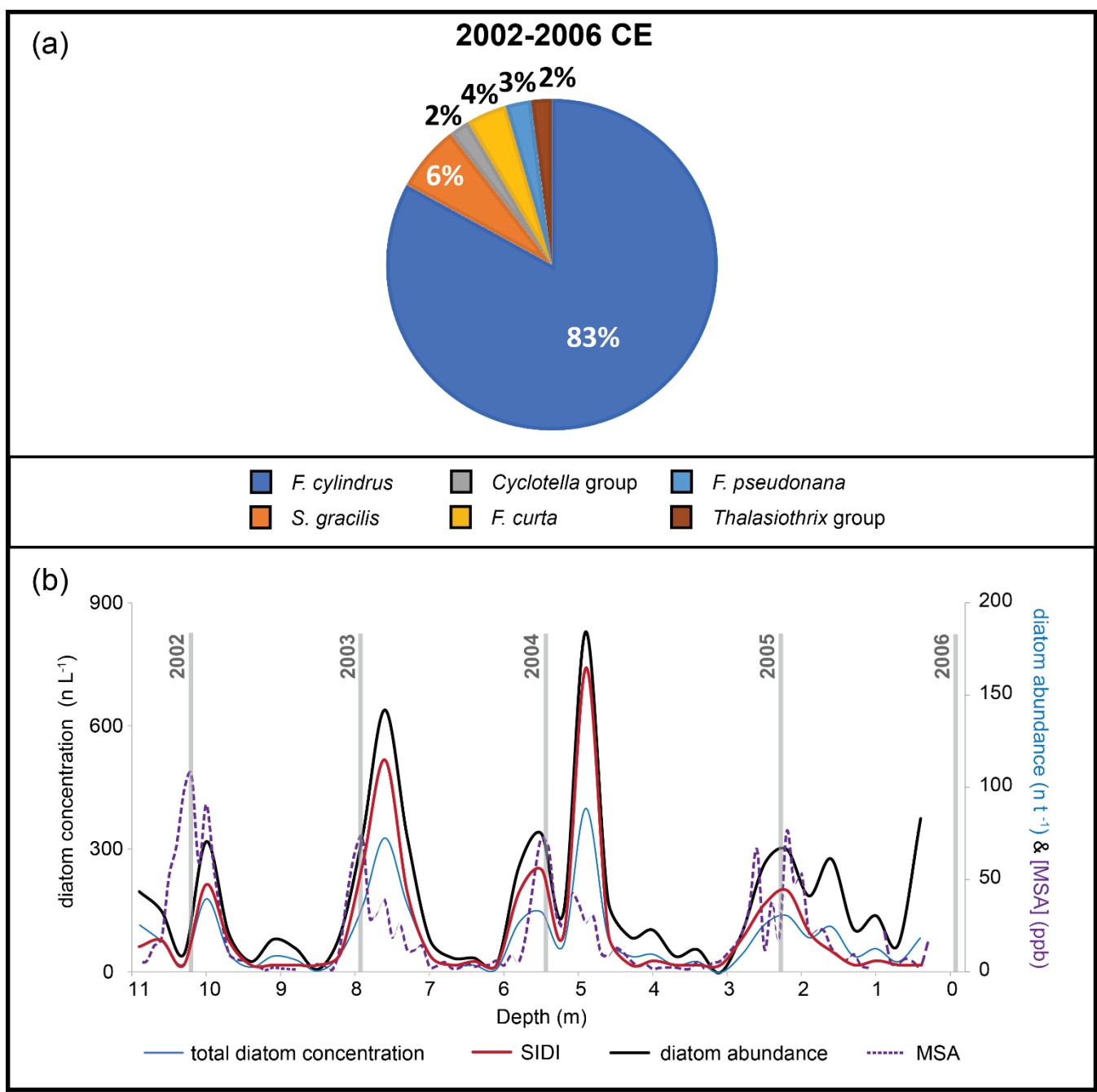

**Figure 5. Diatom record from the ROIC ice core. a) Main diatom assemblage composition of the ROIC ice core (2002-2006 CE). Percentages reported in this figure were normalized to the main species identified. b) Total diatom concentration, total diatom abundance and SIDI variations down-core for the 2002-2006 CE horizon. Vertical grey lines indicate temporal horizons based on the austral summer maxima in MSA.**

## 4.5. Regional diatom ecology

The diatom assemblages at all sites is dominated by *Fragilariopsis* spp. and *Shionodiscus* spp., two genera that are common and abundant in the SO (Crosta et al., 2005; Rigual-Hernandez et al., 2015). An additional group identified in every ice core is the *Cyclotella* group, that is comprised of unspecified specimens of *Cyclotella sensu lato* (including *Lindavia, Discostella, Tertiarius* and *Pantocsekiella* and other morphologically similar types), a cosmopolitan genus-complex with broad ecological affinities across marine, brackish and freshwater environments (Lowe, 1975).

Out of the ten taxa identified in the main diatom assemblages (Table 3), six are exclusively marine, whilst the other four have been identified in marine, brackish and freshwater environments (Lowe, 1975; Van de Vijver and Beyens, 1999; Bouchard et al., 2004; Hamsher et al., 2016; Malviya et al., 2016). The marine taxa include sea ice affiliated diatoms (*F. cylindrus* & *F. curta*) (Zielisnki and Gersonde, 1997; Lizotte, 2001) and open ocean species/groups (*S. gracilis, F. pseudonana, Pseudo-nitzschia* spp. & *Thalassiothrix* gp) (Crosta et al., 2005; Zielisnki and Gersonde, 1997; Rigual-

Hernandez et al., 2015). In total, the marine taxa contribute at least 58% to the assemblages of the four ice core sites and indicate a predominantly marine origin for the diatoms present in the AP and EL ice cores (Figure 6).

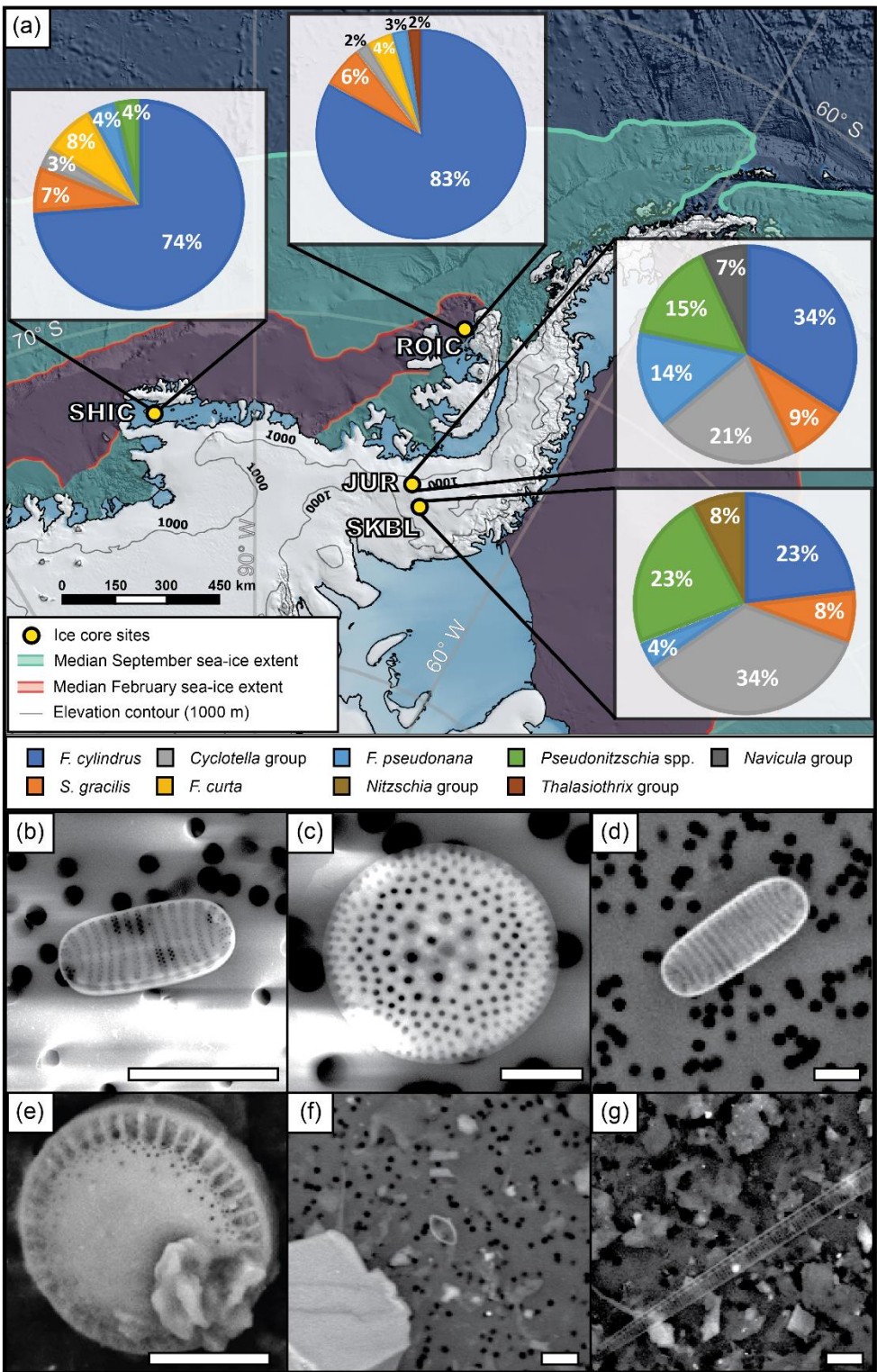

| ■ *F. cylindrus* | ■ *Cyclotella* group | ■ *F. pseudonana* | ■ *Pseudonitzschia* spp. | ■ *Navicula* group |
| ■ *S. gracilis* | ■ *F. curta* | ■ *Nitzschia* group | ■ *Thalasiothrix* group | |

**Figure 6. Main diatom assemblage for each ice core over their overlapping period (2002-2006 CE). a) Map of ice core sites showing the main diatom assemblage for each location. Scanning Electron Microscope (SEM) micrographs of the most abundant diatom taxa b)** *F. cylindrus* **found in ROIC. c)** *S. gracilis* **found in JUR. d)** *F. curta* **found in SHIC. e) Diatom valve as example of the** *Cyclotella* **group found in JUR. f)** *F. pseudonana* **found in JUR. g)** *Pseudonitzschia* **spp. found in SKBL. Scale bar in bottom right of each frame represents five microns.**

## 4.6 Regional distribution

The diatom concentration records from the four ice cores were compared for the overlapping period (2002-2006 CE). The diatom concentration showed a difference between higher mean diatom concentrations at ROIC (178.2 n L$^{-1}$) and SHIC (431.1 n L$^{-1}$) than at JUR (168.3 n L$^{-1}$) and SKBL (56.3 n L$^{-1}$). Diatom concentration in ROIC and SHIC were characterized by higher values (>250 n L$^{-1}$ & >300 n L$^{-1}$ respectively) during austral summer/early autumn, and lower values (<70 n L$^{-1}$ & <100 n L$^{-1}$ respectively) during austral winter/early spring. Conversely, the diatom concentration at JUR and SKBL of 168.3 ± 46 n L$^{-1}$ and 56.3 ± 31.4 n L$^{-1}$, respectively exhibit only minor variations throughout the year and no obvious seasonality.

A regional comparison over the overlapping period (2002-2006 CE) shows the main diatom assemblage also differs across the region. Main diatom assemblages from ROIC and SHIC are dominated by *F. cylindrus* (≥73%) with other species representing minor percentages of the main assemblage. Conversely, JUR and SKBL present three or more species which represent the main proportion of the assemblage. While *F. cylindrus* dominates the assemblage of ROIC and SHIC it contributes ≤34% on JUR and SKBL. The opposite was identified for the *Cyclotella* group where ROIC and SHIC contain ≤3%, while JUR and SKBL contain ≥21%. An additional division was identified in the presence of *F. curta*. This diatom species is widely identified and presents a similar proportion (~4-8%) in ROIC and SHIC, but it is absent in JUR and SKBL.

## 5. Discussion

### 5.1 Diatom source

Regional diatom ecology reveal that the diatom record preserved in ice cores from the Southern AP and EL prevalently conformed by marine taxa abundant in the SO (Crosta et al., 2005; Zielisnki and Gersonde, 1997; Rigual-Hernandez et al., 2015). Marine diatoms have been previously found in numerous ice core sites in Antarctica and their source has been attributed to the SO (Burckle et al., 1988; Kellogg and Kellogg, 1996; Budgeon et al., 2012; Delmonte et al., 2013; Delmonte et al., 2017; Allen et al., 2020; Tetzner et al., 2021a). The marine diatoms analysed in this work were not only well persevered but also present in colonies. The recovery of fresh-looking specimens still articulated in short chains suggests a rapid transport of the cells directly from the source to the ice core sites. These findings support SO surface waters as the principal source of diatoms and aeolian transport as the mechanism to transfer diatoms to the AP and EL ice core sites. This is consistent with previous studies showing airmasses originated in the SO are transported within days to the AP and EL ice core sites (Thomas and Bracegirdle, 2015; Allen et al., 2020). Whilst the SO is the principal source of diatoms to ice cores in

this region, we cannot rule-out contributions from exposed diatom-bearing sediments (most reported from sites within the distally located Transantarctic Mountains (Barrett, 2013)) and fresh/brackish-water bodies. Antarctic non-marine water bodies could potentially contribute with diatoms to the ice core record. Diatoms have been found in low concentrations in Antarctic subglacial, supraglacial and epishelf lakes, and cyoconite holes (Smith et al., 2006; Hodgson et al., 2009; Keskitalo et al., 2013; Stanish et al., 2013). However, these non-marine water bodies are confined, scarce and very localized along the

Amundsen-Bellingshausen coast, all located at low-elevation sites and either seasonally or perennially ice-lidded due to low temperatures year-round (Giralt et al., 2020; Dirscherl et al., 2021). Similarly, cryoconite holes have only been reported in regions distant from the ice core sites, such as the northern tip of the Antarctic Peninsula (South Shetland Islands) (Buda et al., 2020), the McMurdo Dry Valleys in Victoria Land (Fountain et al., 2004; MacDonell et al., 2016; Darcy et al., 2018) and the coast of Dronning Maud Land (Weisleitner et al., 2020), all exposed to the same ice-lidded conditions (Buda et al.,

2020). Therefore, preventing them from becoming major contributors of diatoms to the ice core record and highlighting non-marine diatoms may reflect a minor input from a variety of sources, including intra-continental and lower latitudes. Results presented here show non-marine diatoms have the potential to account for a moderate portion of the diatom assemblage (less than 42%). These results highlight a detailed study of non-marine diatom species can hold valuable environmental information about extra-Antarctic sources. However, in this study, due to the limitations on the identification of non-marine

species, the rest of the discussion included in this section will be centred on the marine portion of the diatom assemblage.

The SO is a vast and diverse region covering major oceanographic zones with varied environmental conditions (Figure 1). Ecological affinities of the marine diatoms present in each ice core indicate the dominant oceanographic source region and suggest that the marine diatoms are principally derived from the SSIZ and the POOZ (See section 2).

The diatom assemblage of SHIC and ROIC are dominated by diatoms associated with the SSIZ (≥68%, *F. cylindrus* and *F.*

*curta*). A prevalent SSIZ source of diatoms for these two sites is also supported by the high mean diatom concentrations and the strong seasonal variability (Table 4), reflecting the typical intense, seasonal blooms that characterise the SSIZ (See section 2). The proximity of the diatom source region to the SHIC and ROIC, and the seasonal opening of coastal polynyas nearby (Arrigo and van Dijken, 2003), may also contribute to the enhanced diatom concentrations at these two sites (Tesson et al., 2016) (Figure 1).

The diatom assemblages of JUR and SKBL are dominated by diatoms associated with the SSIZ and the POOZ (≥58%). Thus, suggesting both the SSIZ (within the SAZ) and the POOZ (within the NAZ) as the source of diatoms for these ice core sites. Despite both oceanographic zones being identified as diatom sources, two lines of evidence support the POOZ as the dominant source region. Compared with the ROIC and SHIC coastal ice cores, JUR and SKBL contain a lower proportion of sea ice diatoms (<34% & ≤23.1% respectively), suggesting reduced transport from the SSIZ. The comparatively higher

proportion of the more distally-sourced, open ocean diatoms denote greater transport from the NAZ (Figure 1). The sub-annual samples also support the POOZ within the NAZ as the main diatom source. The lack of seasonality detected in the JUR and SKBL sub-annual diatom records is consistent with the modest seasonality in primary production observed in the NAZ of the Pacific sector (Arrigo et al., 2008; Soppa et al., 2016). Moreover, the reduced concentration of sea ice diatoms

(*F. cylindrus & F. curta*) and its lack of correlation with the variability of the total diatom concentration, suggests the SSIZ plays a modest role in shaping the diatom record at these two sites (Figure 2c and Figure 3c). Overall, the JUR and SKBL diatom records indicate the POOZ (within the NAZ) as the primary source of diatoms to these sites, with limited contributions from the SSIZ.

The different source regions for the ROIC and SHIC versus the JUR and SKBL diatom records, is likely due to their locations. ROIC and SHIC are coastal, low elevation sites whilst JUR and SKBL are inland, high elevation sites. Back trajectory analyses reveal that airmasses arriving at high elevation sites (JUR and SKBL) are in contact with the sea surface farther offshore, north of the SSIZ and therefore entrain mostly open ocean diatoms (Thomas and Bracegirdle, 2009; Thomas and Bracegirdle, 2015; Allen et al., 2020).

The identification of two different diatom source regions for ice core sites located in contrasting geographical locations is consistent with previous findings across Antarctica. A SSIZ source for coastal regions is consistent with previous findings from a coastal site (~400 m a.s.l, 10 km away from the coast and 50 km from the ice-free ocean) in Windmill Island, East Antarctica (Budgeon et al., 2012). At this site, diatom concentrations ranged from 0-180 (n $L^{-1}$) and the diatom main assemblage was almost exclusively composed of *F. cylindrus*, *F. curta*, *S. gracilis* and *F. pseudonana*. Similarly, our results from inland sites agree with the results obtained from the Ferrigno ice core, drilled at an inland location in EL (1354 m a.s.l., 140 km away from the coast) (Allen et al., 2020). At this site, an open water region within the NAZ was identified as the dominant diatom source and diatom concentration values (0-140 n $L^{-1}$) were comparable to the values obtained for JUR and SKBL. The prevalence of marine diatoms preserved in the records from inland high-elevation sites in the AP and EL region (JUR, SKBL and FER) contrasts with the predominance of freshwater and reworked diatoms previously recorded in Antarctic ice cores from continental sites such as South Pole, Dome C and Vostok (Burckle et al., 1988; Kellogg and Kellogg, 1996; Kellogg and Kellogg, 2005). This disparity shows ice cores from the AP-EL region are ideally situated to entrain Antarctic marine taxa as a large proportion of the diatom assemblage.

## 5.2 Inter-annual variability

A first step in understanding the temporal variability in the diatom record is to examine the relative role of ice core site conditions and post-depositional processes. The lack of correlation between the diatom abundance, meltwater volume and ice core snow accumulation suggest that deposition of diatoms occurs under a mixed regime, which does not depend on precipitation changes at the ice core site. Similarly, no clear relationship was identified between the sub-annual diatom abundance and the monthly mean wind speed measured at JUR, SKBL and in the vicinities of ROIC (Thomas and Bracegirdle, 2015; Tetzner et al., 2019). These results demonstrate that the magnitude and variability of the diatom records are not controlled by local environmental conditions at the ice core sites.

Similar results have been previously reported for the Ferrigno ice core site, where the annual diatom abundance presented a weak and non-significant correlation with the volume of meltwater filtered per sample (R=0.14, p>0.05) and with the annual snow accumulation (R=0.12, p>0.05) (Allen et al., 2020). Our results, and the results presented for the Ferrigno ice core,

contrast with the depositional mechanisms of insoluble mineral dust over the ice sheets (Sudarchikova et al., 2015). In particular, insoluble mineral dust has been shown to be deposited either via wet (snow scavenging in the atmosphere) (Wolff et al., 1998; Breider et al., 2014), dry/wet (Koffman et al., 2014) or dry deposition (gravitational settling) (Li et al., 2010),

depending on the location of the ice core site. Likewise, deposition of insoluble mineral dust has been shown to be enhanced under weak wind conditions, which favours the gravitational settling of dense particles (Fernandes et al., 2019). Both observations contrast with our results which show diatoms are not deposited under specific wind or precipitation regimes, possibly due to their high surface-area:mass ratio (Scherer et al., 2016), enabling them to stay afloat.

Whilst the potential effects of post-depositional processes such as snow ablation and redeposition cannot be ruled out

(Lenaerts and Van den Broeke, 2012, van Wessem et al., 2016), the continuity and regularity seen in the $H_2O_2$ seasonal cycle indicate that these ice core records were not disrupted by major ablation or redeposition events. Altogether, results presented in this work reveal that local environmental changes are not the main drivers of the temporal variability in the diatom record preserved in the AP & EL ice cores.

Decadal subset analyses of the diatom concentration revealed a regional increase of 41.39 %, 25.56 % and 63.76 % for JUR

(1992-2012 CE), SKBL (1999-2019 CE) and SHIC (1999-2019 CE) respectively, between the first and second decades. The consistent increase in diatom concentrations over the three sites suggests there may be a common driver of the temporal variability in the diatom record. Firn compaction affects every ice core site regardless of their location. Even though the continuous deposition of snow on the surface adds a progressive load on top of the diatoms preserved in deeper ice core layers, diatom frustules have shown to withstand pressures equivalent to 700 tonnes $m^{-2}$ without fracturing (Hamm et al.,

2003). Similarly, the stable proportion of fragments relative to diatom frustules, and pennate fragments relative to centric fragments, down-core, supports diatoms are not preferentially fractured by the ice load. Moreover, the recovery of fresh-looking specimens still articulated in short chains and preserving delicate ornamentation at the bottom of these ice cores evidence the diatom records were not affected by mechanical fracturing or chemical dissolution downcore. Thus, proving the recent increase in the diatom concentration is not caused by post-depositional processes progressively affecting the diatom

record down-core. In turn, these evidence supports diatom fragments are not produced within the ice, highlighting they are fragmented before their deposition, either while transported in the atmosphere (Marks et al., 2019), while suspended in subaquatic environments (Gersonde and Wefer, 1987; Budgeon et al., 2013), and/or after being exposed to subaerial environments (McKay et al., 2008).

Decadal subset analyses of the diatom assemblages revealed only minor variations in composition (Figures 2a, 3a and 4a),

confirming that the principal sources (POOZ and SSIZ) have remained stable over the last two-to-three decades. Since the diatom source areas have not moved, the recent increase in diatom concentration likely reflects environmental changes within the POOZ and SSIZ of the ABS (and/or transport efficiency). The SO/ABS has recently experienced sustained and considerable changes in atmospheric circulation and sea ice dynamics over recent decades (See section 2). The POOZ is located within the SWW belt and therefore prone to be affected by changes in the strength and position of the SWW

(Mayewski et al., 2013; Menviel et al., 2018). Recent strengthening and southern shift in SWW (Goyal et al., 2021) as the

potential driver of the increased diatom concentrations observed in JUR and SKBL is consistent with the strong correlation between the ice core diatom record and changes in wind strength over the SO reported by Allen et al. (2020). For the SHIC, the close link between the diatom record and the local SSIZ conditions (Arrigo et al., 2008; Arrigo et al., 2012) suggest that variations in the ABS SSIZ will be reflected in the SHIC diatom record. In particular, the recent sustained decrease in the area of the ABS SSIZ (Parkinson, 2019) has shortened the distance between the SSIZ and the ice core sites. Similarly, the prolonged ice-free season (Stammerjohn et al., 2012) has extended the exposure of stratified waters in the SSIZ. Both, potentially increasing the availability of diatoms to be transported to SHIC. Altogether, the recent increase in diatom concentration across the region likely reflects the observed environmental changes within the POOZ and SSIZ.

## 6 Conclusions

Diatoms are faithful recorders of environmental conditions. Resolving the environmental controls on the assemblage and abundance variations of diatoms in different Antarctic ice cores offers the potential to establish a new and unique paleoenvironmental proxy. Our multi-site assessment of diatoms preserved in Antarctic Peninsula and Ellsworth Land ice cores confirm that the 20 year record is mainly comprised by pristine specimens of Southern Ocean marine diatoms. Diatoms in the two coastal ice cores are largely comprised of sea ice taxa and exhibit consistent timing of peak inputs during austral summer. At inland sites, the diatom records are characterized by open ocean species with relatively constant inputs throughout the year. This strong geographical division can be exploited to recover valuable environmental information from both the sea ice and open ocean regions.

Diatom records from all four Antarctic Peninsula and Ellsworth Land ice cores reveal a recent rise in diatom concentrations. We demonstrate that this regional increase is not driven by changes in local conditions at the ice core sites or in the diatom sources, but is likely a result of stronger winds entraining and transporting more diatoms and/or declining sea ice extent reducing the transport distance. Altogether, our findings emphasize how the diatom record preserved in Antarctic ice cores has the potential to become a robust proxy of environmental conditions in the Southern Ocean.

The strong seasonality of the diatom record at coastal sites also holds potential as a new chronological marker, providing a novel tool to date ice cores where the effects of climate change (e.g. Surface melt, increased rain events) impair traditional annual layer counting (Simoes et al., 2004; Fernandoy et al., 2018; Thomas et al., 2021).

Overall, the evidence presented here confirms that diatoms preserved in ice cores from the Antarctic Peninsula and Ellsworth Land yield robust, regionally consistent records with the potential to deliver novel environmental proxies and a new chronological tool. Further research should be focused on exploring the spatial relation between the diatom record and environmental parameters.

## Data availability

Datasets original to this work will be available at the UK Polar Data Center (https://www.bas.ac.uk/data/uk-pdc/).

## Author contribution

DT did the initial conceptualization. DT, CA and ET conducted the formal analysis. DT was in charge of the Investigation. DT and CA designed the Methodology. DT prepared the original manuscript. CA and ET contributed to the reviewing and editing of the original manuscript.

## Competing interests

The authors declare that they have no conflict of interest.

## Acknowledgements

We would like to thank Sarah Crowsley, Tom King, Isobel Rowell and Dr Robert Mulvaney for their help while drilling the SHIC and SKBL ice cores included in this work. We would like to thank Shaun Miller, Dr Jack Humby, Dr Diana Vladimirova and Dr Daniel Emanuelsson from the Ice core Lab, British Antarctic Survey, for their help while cutting the ice and conducting the Continuous Flow Analysis (CFA). We would like to thank Professor Eric Wolff from the Earth Sciences Department, University of Cambridge, for his comments during the final review and editing of this draft. SEM work was partly supported by a Royal Society Research Professorships Enhancement Award (RP\EA\180006). We would like to thank Dr Iris Buisman and Dr Giulio Lampronti from the Microscopy Lab, Earth Sciences Department, University of Cambridge, for their technical support in the use of the SEM. Fieldwork conducted for this research was supported by the Collaborative Antarctic Science Scheme (CASS-168) This research was funded by CONICYT–Becas Chile and Cambridge Trust funding program for PhD studies. Grant number 72180432. Finally, the authors thank Professor Viv Jones, an anonymous reviewer and the editor for their constructive comments that led to an improved manuscript.

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
