# Peer review of "Regional variability of diatoms in ice cores from the Antarctic Peninsula and Ellsworth Land, Antarctica"

_The Cryosphere, 2021_

## Referee Comment (RC1)

In this paper diatoms have been extracted from 4 ice cores from Antarctica covering the last *c.* 20 years and have been enumerated at annual and sub-annual resolution. Coastal and inland sites differ in the proportions of sea ice taxa and those from open marine environments, and there is a regional increase in the concentration of diatoms this is information that can be useful for environmental reconstruction.

The paper is clearly articulated and well-written and contains original interesting data making an important contribution to the literature. I have a couple of major points of clarification and discussion.

**Non-marine diatoms**
The presence of non-marine diatoms needs some further discussion throughout the paper.

Firstly, in the introduction. Here there needs to be more of an acknowledgement that diatoms are also abundant and diverse in lakes, streams and wet habitats (e.g. soils) in terrestrial environments of Antarctica as well as growing on ice e.g. in cryoconite holes (Verleyen et al 2021, Noga et al 2020, Cavacini 2001, Van de Vijver & Beyens 1998). Thus there are additional sources of diatoms which either can grow *in situ* or be blown onto the ice which need to be taken into consideration. The authors mention some of these sources later in the paper, but they need to be referred to in the introduction. Diatoms in cryoconites aren't mentioned at all, are these environments present in their sampled locations? could they contribute an additional source?

Secondly, the 'broad ecological affinities' of the *Cyclotella* group (L300) needs a bit more unpacking. Many of these species are associated with freshwaters; the *Cyclotella* group as defined here may be considered to have a broad affinity but within it specific taxa have much narrower niches. By grouping together in this way information has been lost. Given the authors' comments about good preservation (e.g L179) it would have been useful to know (perhaps in the methods) the constraints on greater taxonomic resolution. Why was it not possible to identify diatoms to species level? If species had been determined then these could be more confidently assigned to aerial, freshwater, brackish or marine categories – diatom species are pretty faithful, especially in terms of salinity. *Cyclotella* is an important component at ice cores JUR and SKBL so there needs to be a bit more discussion of the possible genera and species found in the ice cores and where these are found in present day communities.

The same point applies to the *Navicula* and *Achnanthes* groups (Table 3) as species from these groups/genera can be found in aerial or freshwater habitats as well as marine ones.

The fact that marine diatoms consist of just over half (58%) of the assemblages (L307, Table 3) at the 4 sites means you do have to spend a bit of time thinking about diatoms from non-marine environments too. It is misleading to say in terms of marine components that (L330) they are – 'almost exclusively dominated' or (L341) there are 'minor contributions' from exposed sediments and freshwater and brackish. Stating (L380) that ice cores from this region are uniquely situated to record marine transported diatoms over emphasises this importance of the marine diatoms.

Perhaps this is a missed opportunity? it's not binary i.e., an ice core could provide useful environmental interpretations from both marine and terrestrial sources if those sources can be well separated which they should be if high enough taxonomic resolution is employed. For example, questions re. extent of terrestrial sources could also be explored.

In sections 2.1 2.2 and 2.3 current conditions in terms of oceanographic, climate and sea ice are detailed but there is no mention of what is happening on the land...if we accept that there

is, or at least could be, a terrestrial signal in the ice cores then a brief discussion of recent terrestrial changes would be justified.

**"Unclassified' Diatom data**

In Table 2 high proportions of diatoms are 'unclassified' e.g. in JUR 544 diatoms out of 1149 (i.e. *c.* 50%) but also high at other sites (e.g. SHIC *c.* 25%), I assume these are the (L157) "unidentified (obscured, undiagnostic or indistinct)" diatoms. With so many in this category (I acknowledge when doing diatom analysis allocating some diatoms to this category is inevitable, but in my experience, these would be unusually high proportions) there needs to be a bit more detail as to why and whether this has any consequences. For example, if they are obscured then by what? What is meant by undiagnostic? And indistinct? Most diatoms can be identified to species level (or if not identified with confidence to a described species at least put into separate taxonomic units) so is this an issue with the technique, for example the use of SEM rather than light microscope? Some comments would be useful. What I am worried about here is some sort of systematic bias, for example if some genera are more undiagnostic than others due to being say harder to identify.

The percentage diatom assemblage data presented in Figures (2 -4) are "normalised" to the main species (i.e. recalculated to 100%) but when you have varying amounts of unidentified species it makes it difficult to compare between sites. Ideally, you need to look at the whole assemblage if not then there is an assumption that all the unidentified diatoms are a random subsample (not biased) of the assemblage? Is that a reasonable assumption?

I assume that as with light microscopy fragments are counted e.g. a fragment of a raphid diatom central area is usually distinct enough to assign to a genus if not a species. But if fragments form part of the 'indistinct' category then there is a need to square this with the pristine nature (ornamentation and chains are mentioned L409) of the diatoms. Again, this could be covered with more detail on what constitutes the 'unclassified' portion.

**Minor issues**

Fig 1 the locations of the ice cores aren't particularly clear – an inset would help here (like that of Figure 6..or just refer to that) and the caption needs to include the names of the ice cores. Locations such as Amundsen-Bellinghausen seas, Ellsworth Land need to be added.

What is the justification of the decadal subsets? if these are "to assess the consistency of the assemblage" (L160) there might be better ways to do that e.g. DCA axis scores to see if there are changes in 'turnover'. Subsequently there is more emphasis on using the decadal data sets to examine changes in diatom concentration (number per L) between the 2 time periods (L404) and these are compared directly but the time intervals aren't the same e.g. JUR is 1992-2001 and 2002-2012 whilst SHBL and SHIC are 1999-2008 and 2009-2019. So I am unclear as to the justification for these decadal means when all 3 cores show positive trends over the whole sampled period and section 2 doesn't discuss climate/sea ice changes at these time intervals.

Line 267 S. gracilis (not gracilics)

**References**

Verleyen, E., Van de Vijver, B., Tytgat, B., Pinseel, E., Hodgson, D. A., Kopalová, K., Chown, S. L., Van Ranst, E., Imura, S., Kudoh, S., Van Nieuwenhuyze, W., consortium, A., Sabbe, K., & Vyverman, W. (2021). Diatoms define a novel freshwater biogeography of the Antarctic. *Ecography*, *44*(4), 548–560. https://doi.org/https://doi.org/10.1111/ecog.05374

Noga T, Kochman-Kędziora N, Olech M, Van de Vijver B. Limno-terrestrial diatom flora in two stream valleys near Arctowski Station, King George Island, Antarctica. Polish Polar Research. 2020:289-314.

Cavacini P. 2001. Soil algae from northern Victoria Land (Antarctica). Polar Bioscience 14: 45–60.

Van de Vijver & Beyens1998. A preliminary study on the soil diatom assemblages from Ile de la Possesion (Crozet, Subantarctica). European Journal of Soil Biology 34:133–141.

Stanish, L. F., Bagshaw, E. A., McKnight, D. M., Fountain, A. G., & Tranter, M. (2013). Environmental factors influencing diatom communities in Antarctic cryoconite holes. *Environmental Research Letters*, *8*(4), 45006. https://doi.org/10.1088/1748-9326/8/4/045006

---

## Author Response (AR1)

Dear Editor,

We appreciate all your comments and suggestions. We also appreciate the comments and suggestions from Professor V. Jones and the Anonymous Reviewer. We thank you all for your time and consideration reviewing our manuscript. Please find to follow an updated list of all points raised, our updated responses to each item and where to find our modifications in the revised manuscript.

**Editor comments**

**Reviewers comment #1**

Line 12: erase southern

Lines 16-17: replace for "Antarctica"

**Response:**

Modified as suggested (Lines 11, Lines 15)

**Reviewers comment #2**

Line 22: you could add 2 aspects here: 1- that because diatoms are dense, they are particulalry good at promoting carbon export from surface to deep waters. 2- the silicified/hard nature of the particles allows preservation in sediments and ice, making them a valuable paleo proxy.

**Response:**

Modified as suggested (Lines 22-24)

**Reviewers comment #3**

Line 28: could be worth adding something around size. ie. small diatoms may be transported by wind further away from their sources compared to larger diatoms.

**Response:**

Unlike dust grains and other solid particles, diatoms have a high surface-area:mass ratio. This feature makes diatoms likely to be transported for long distances, regardless of their size. We appreciate your suggestion, however, we will not include it as there is not conclusive evidence about the statement.

**Reviewers comment #4**

not by m2. the North Atlantic or North Pacific, or coastal waters, have higher productivity/m2 than the in Southern Ocean. Most of it is actually low PP relative to the amount of Nutrients available.

**Response:**

We agree with your comment. However, in our manuscript we state "The SO is one of the most productive water masses on Earth". We do not intend to claim that the SO is the most productive water mass on Earth.
* * *
**Reviewers comment #5**

Legend Figure 1: Modify "extension" for "extent"

**Response:**

Modified as suggested (Legend Figure 1 and Figure 6)
* * *
**Reviewers comment #6**

Line 89 and 95: modify XXth century

**Response:**

Modified as suggested (Lines 99 and 105)
* * *
**Reviewers comment #7**

Lines 133, 137 and 138: replace hydrogen peroxide for H2O2

**Response:**

Modified as suggested (Lines 156, 160, 161).
* * *
**Reviewers comment #8**

Line 160: change "20 year" for "20-year"

**Response:**

Modified as suggested (Line 187).
* * *
**Reviewers comment #9**

Line 172: change "extension" for "extent"

**Response:**

Modified as suggested (Line 199).
* * *
**Reviewers comment #10**

Line 183: is it possible to add size class in the table?

**Response:**

Individual diatom size measurements were not performed while analysing the samples. We hope to explore the regional variability of diatom sizes in subsequent work.

**Anonymous reviewer's comments**

**Reviewers comment #1**

Figure 5 and MSA: It is mentioned in the methods that the ROIC ice core was dated using the annual cycles of major ion concentrations, and MSA is used in Figure 5c to identify the austral summer based on the maxima of MSA. As this study crosses multiple disciplines, not every reader will know what MSA is and understand its use here. MSA should be introduced earlier on in the manuscript (only mention is line 129) as to what it is, how it is interpreted at this site, and what the maximum represents.

**Response:**

Revised as suggested. Details about MSA and how it can be interpreted are now included in the methods section (Lines 150-153).

**Reviewers comment #2**

This may be a missed opportunity in the discussion for ROIC site, as MSA can be an indicator of winter sea ice extent or summer primary productivity (i.e., Thomas et al., 2019 as cited; Abram et al., 2013), and the relative timing of the MSA peak to the diatom peaks could add an interesting aspect to the discussion.

**Response:**

The link between MSA and diatoms in ice cores is not straightforward. Despite MSA being an indicator of primary productivity, this does not necessarily imply a direct relationship between the concentration of MSA and the diatom abundance in ice cores. To date, there is uncertainty on the amount of DMS (turned into MSA after oxidized) produced by each diatom species. We acknowledge that exploring the relationship between MSA and diatom abundance could provide interesting insights but it is beyond the scope of this research article. We hope to explore the relationship between MSA and ice core diatom assemblages in subsequent work.

**Reviewers comment #3**

Line 331/376: The text emphasizes the 'dominance' of marine diatoms in the record, when they are just over 58% (line 307) in some instances. The authors also state they cannot "rule out minor contributions from exposed sediments in fresh/brackish-water bodies." in line 341. Given that in some instances marine diatoms only make a little over half the assemblages I suggest the authors rephrase these sentences as 'dominance' could overemphasize the marine contribution. The same could be said about 'minor' minimizing contributions from fresh/brackish water bodies.

**Response:**

Marine diatoms account for at least 58% of the main diatom assemblage preserved in ice cores presented in this work. We have modified the introduction and discussion sections, accounting for the potential inputs from non-marine sources. We have also revised the text to better acknowledge the contribution of "non- marine" taxa. Finally, we amended our description of the Antarctic marine taxa as 'prevalent' within the diatom assemblage rather than 'dominant' throughout the whole manuscript (Lines 365-366, 374-376, 426, 18-20)

**Reviewers comment #4**

Line 344: SSIZ and POOZ are used to describe 'seasonal sea ice zone' and 'permanently open ocean zone' with a reference to section 2. However, neither of these acronyms are used or defined in section 2. SSIZ is defined in the caption of Figure 1. POOZ isn't mentioned until the caption in Table 3. As these acronyms are used extensively in the discussion, they should be defined and described in main text in section 2 and not just in the captions.

**Response:**

Modified as suggested (Lines 59-64).

**Reviewers comment #5**

Section 5.2 Inter-annual variability: This section provides some interesting insights, but the assessment of changing environmental conditions influencing diatom concentrations could be strengthened (from line 413 onwards). First off, regarding the decadal variability. Why were these subsets chosen? As they differ between sites, they seem arbitrary. The discussion could instead focus on the overall increasing trend, rather than the differences between the subsets, particularly as the discussion of the changes in atmospheric circulation and sea ice dynamics is only regarding 'recent decades' and not these specific periods. However, since these data were subdivided, it would strengthen the discussion to be more specific about the trends in environmental conditions over each decade analysed. For example in Line 423: Regarding the 'recent decrease in the area of the ABS SSIZ mentioned in Parkinson, 2019– the overall trend is negative, but there has been a slight increasing trend since ~2010. This would then perhaps suggest an increase in the distance of the SSIZ relative to the 1999-2008 period. While data for the specific decades analysed may not be readily available for all environmental conditions mentioned in the text, the authors should make improvements where possible.

**Response #5:**

Our reason to analyse temporal changes in the assemblages and on the diatom concentration was to assess the consistency of the diatom record in response to recent environmental changes in the

region. In particular, to determine if there were shorter-term shifts in the diatom concentrations caused by changes in certain diatom species or if there was a general shift in all diatom species that shape the main assemblage. We decided to assess this at a decadal timeframe to reduce the potential imprints of interannual variability. We did not discuss specific changes in climate/sea ice during the assessed decades because trends in both, wind strengthening and sea ice retreat, have been sustained over the last decades.

We acknowledge we did not specify that wind and sea ice trends have been sustained over time or refer to the different timing of the decadal subsets across the sites. To address this comment, we have modified the manuscript, specifying our reasons to analyse the dataset in decadal subsets (Lines 189-191), we have clarified that recent trends have been sustained over the last decades (Lines 471 and 479) and we specified the timeframes considered when comparing recent changes in the diatom concentration (Lines 454-455).
* * *
**Reviewers comment #6**

Abstract, line 16: "yield a novel wind paleoenvironmental proxy" – suggest 'paleoenvironmental proxy' as authors acknowledge other environmental factors may influence diatom content in ice cores such as sea ice extent.

**Response:**

Modified as suggested (Line 15).
* * *
**Reviewers comment #7**

Line 28: "over long distances" – if possible, be more specific here (ex: over XX kms)

**Response:**

Specified as requested (Lines 29-30).
* * *
**Reviewers comment #8**

Line 44: "ocean" is capitalized

**Response:**

Modified as suggested (Line 47).
* * *
**Reviewers comment #9**

Figure 1: SSIE and PSIE are difficult to read/see in the main figure – perhaps outline these in white (but keep them filled in with color) to make them stand out more?

**Response:**

Modified as suggested (Figure 1).

**Reviewers comment #10**

Coastal polynyas: The transect in Figure 1 identifies coastal polynyas as a feature in this area. There are several in this region, yet no mention of coastal polynyas is made in the text or how polynya variability may impact these records.

**Response:**

This paper aims to determine the regional and temporal variability of diatom records preserved in ice cores from the AP and EL regions. Results presented in this paper allow us to broadly outline diatom sources and suggest potential processes driving the variability of the record. We agree polynyas in this region could potentially contribute diatoms to ice core sites. We have included this in our revised manuscript (Lines 397-399).
* * *
**Reviewers comment #11**

Lines 89 and 95: "XXth century"

**Response:**

Modified as suggested (Lines 99 and 105).
* * *
**Reviewers comment #12**

Table 1 caption: When describing SIE – perhaps refer to section 3.3 which provides the source of the data. The caption does an excellent job of explaining how these distances were calculated but does not identify the source of the data which is presented later.

**Response**

Modified as suggested (Caption – Table 1)
* * *
**Reviewers comment #13**

Figures 2-5, Part C: The color used for SIDI can be hard to differentiate from the total diatom concentration (particularly with a printed version)– an alternate color or line marker may be more suitable. However, this is up to the author and editor's discretion.

**Response**

Figures 2, 3, 4 and 5 were modified as it was suggested. (Figures 2-5)

**Professor V. Jones comments**

**Prof V. Jones comment #1**

The presence of non-marine diatoms needs some further discussion throughout the paper.

Firstly, in the introduction. Here there needs to be more of an acknowledgement that diatoms are also abundant and diverse in lakes, streams and wet habitats (e.g. soils) in terrestrial environments of Antarctica as well as growing on ice e.g. in cryoconite holes (Verleyen et al 2021, Noga et al 2020, Cavacini 2001, Van de Vijver & Beyens 1998). Thus there are additional sources of diatoms which either can grow in situ or be blown onto the ice which need to be taken into consideration. The authors mention some of these sources later in the paper, but they need to be referred to in the introduction.

**Response:**

Additional information about other environments where diatoms can be found was added in the Introduction as suggested (Lines 18-20).

**Prof V. Jones comment #2**

Diatoms in cryoconites aren't mentioned at all, are these environments present in their sampled locations? could they contribute an additional source?

**Response:**

Cryoconite holes are common in Alpine and Artic glaciers. However, in Antarctica, cryoconite holes have been comparatively rare, with most of them located in the McMurdo Dry Valleys (**Fountain et al., 2004**), some in the Dronning Maud Land Coast, East Antarctica (**Weisleitner et al., 2020 and references therein**) and just one report of cryoconite holes from the northern AP, in the South Sandwich Islands (**Buda et al., 2020**). Unlike cryoconite holes in Alpine and Artic Glaciers, Antarctic cryoconite holes are mostly perennially ice-lidded due to low air temperatures year-round, so it is unlikely that they are a prevalent source of diatoms to the ice core records presented in this study. We have addressed this comment by acknowledging cryoconite holes as potential source of diatoms (Lines 18-20) and by describing the scarcity of cryoconite holes (and other non-marine water bodies) in this region and their limitations to become a common source of diatoms to our ice core sites (Lines 381-386).

**References:**

Buda, J., Łokas, E., Pietryka, M., Richter, D., Magowski, W., Iakovenko, N. S., ... & Zawierucha, K. (2020). Biotope and biocenosis of cryoconite hole ecosystems on Ecology Glacier in the maritime Antarctic. Science of the Total Environment, 724, 138112.

Fountain, A. G., Tranter, M., Nylen, T. H., Lewis, K. J., & Mueller, D. R. (2004). Evolution of cryoconite holes and their contribution to meltwater runoff from glaciers in the McMurdo Dry Valleys, Antarctica. Journal of Glaciology, 50(168), 35-45.

Weisleitner, K., Perras, A. K., Unterberger, S. H., Moissl-Eichinger, C., Andersen, D. T., & Sattler, B. (2020). Cryoconite hole location in East-Antarctic Untersee Oasis shapes physical and biological diversity. Frontiers in Microbiology, 11, 1165.
* * *
**Prof V. Jones comment #3**

Secondly, the 'broad ecological affinities' of the *Cyclotella* group (L300) needs a bit more unpacking. Many of these species are associated with freshwaters; the *Cyclotella* group as defined here may be considered to have a broad affinity but within it specific taxa have much narrower niches. By grouping together in this way information has been lost. Given the authors' comments about good preservation (e.g L179) it would have been useful to know (perhaps in the methods) the constraints on greater taxonomic resolution. Why was it not possible to identify diatoms to species level? If species had been determined then these could be more confidently assigned to aerial, freshwater, brackish or marine categories – diatom species are pretty faithful, especially in terms of salinity. *Cyclotella* is an important component at ice cores JUR and SKBL so there needs to be a bit more discussion of the possible genera and species found in the ice cores and where these are found in present day communities. The same point applies to the *Navicula* and *Achnanthes* groups (Table 3) as species from these groups/genera can be found in aerial or freshwater habitats as well as marine ones.

**Response:**

The reason to combine all diatoms identified as *Cyclotella* into a single *Cyclotella* group was due to the wide variety of diatom frustules with recognisable *Cyclotella*-like features but unidentified to species level due to insufficient image resolution, partial obscurance, or absence of diagnostic features. Filters were analysed for the presence of diatoms using an SEM equipped with an automated stage that acquires a grid of pictures with a dynamic focusing interpolation. The automated system images the filters in an efficient and optimized way. However, it does not permit zooming-in and re-focusing on individual diatoms to get a higher-definition image to differentiate subtle differences in the ornamentation. We circulated the best images of the most common *Cyclotella*-types to several colleagues in the polar diatom community and received only one suggested identification and several comments on being unable to resolve diagnostic features. Whilst we can't be sure of the source of these taxa, we can be confident that they are not sourced from the sea ice zone or open ocean south of the Polar Front, where these morphologies have not been recorded in either water column or sediment samples. Because of these limitations, we decided to group all *Cyclotella*-type morphologies into one group (The same applies to *Navicula* gp. and *Achnanthes* gp). We acknowledge a detailed classification of diatom species would yield additional information to better understand the diatom record preserved in ice cores and is something that we intend to explore and resolve in future work.

To address this comment, we have added further information on the constraints of our classification in the method section (Lines 175-185) and amended our description of the Antarctic marine taxa as 'prevalent' within the diatom assemblage rather than 'dominant' (across the whole manuscript).
* * *
**Prof V. Jones comment #4**

The fact that marine diatoms consist of just over half (58%) of the assemblages (L307, Table 3) at the 4 sites means you do have to spend a bit of time thinking about diatoms from non-marine environments too. It is misleading to say in terms of marine components that (L330) they are – 'almost exclusively dominated' or (L341) there are 'minor contributions' from exposed sediments and

freshwater and brackish. Stating (L380) that ice cores from this region are uniquely situated to record marine transported diatoms over emphasises this importance of the marine diatoms.

Perhaps this is a missed opportunity? it's not binary i.e., an ice core could provide useful environmental interpretations from both marine and terrestrial sources if those sources can be well separated which they should be if high enough taxonomic resolution is employed. For example, questions re. extent of terrestrial sources could also be explored.

**Response:**

We absolutely concur that the non-marine taxa offer the potential for greater environmental understanding and intend to explore this in subsequent work. However, a previous study has already pointed out the numerous limitations to identify the source(s) for non-marine diatoms in Antarctic snow air and snow samples (**McKay et al., 2008**). Since considerable time investment would be needed (manually locating and re-imaging individual specimens, undertaking several taxonomic investigations, etc.) with no certainty that definitive source region(s) could be identified, we consider this to be beyond the scope of this work.

As reported in our results section, at least 58% of the classified diatoms from each site were exclusively marine. These correspond to all the unambiguous classifications. The remaining fraction, the ambiguous group, comprised specimens identified as benthic, brackish and/or freshwater (some could also be marine e.g. *Navicula*) and diatoms that were, at best classified to genus level (excluding exclusively marine genus).

The broad ecological affinities of these groups prevented us from establishing a clear origin for the "non-exclusively marine" fraction of the assemblage. We acknowledge there is a possibility that the whole "non-exclusively marine" fraction of the assemblage could be entirely comprised by non-marine diatoms. To address this comment, we have modified the discussion section, accounting for the potential inputs from non-marine sources (Lines 374-390).

As previously mentioned in Response #3, we acknowledge a detailed classification of diatom species that comprise our diatom groups would contribute valuable information to better identify their sources and intend to complete this as part of future research. In the manuscript we have added a clear statement of our intention to focus on the significance of the Antarctic marine taxa contribution to the assemblage in this study and acknowledge the similar potential in identifying the remaining taxa and their source regions. (Lines 386-390).

We also amended the text to better acknowledge the contribution of "non- marine" taxa.

**Reference**

McKay, R. M., Barrett, P. J., Harper, M. A., & Hannah, M. J. (2008). Atmospheric transport and concentration of diatoms in surficial and glacial sediments of the Allan Hills, Transantarctic Mountains. Palaeogeography, Palaeoclimatology, Palaeoecology, 260(1-2), 168-183.
* * *
**Prof V. Jones comment #5**

In sections 2.1 2.2 and 2.3 current conditions in terms of oceanographic, climate and sea ice are detailed but there is no mention of what is happening on the land...if we accept that there is, or at least could be, a terrestrial signal in the ice cores then a brief discussion of recent terrestrial changes would be justified.

**Response:**

We added a paragraph in section 2 ("Regional Settings") presenting a brief review of recent terrestrial changes in the region (Lines 115-123).
* * *
**Prof V. Jones comment #6**

In Table 2 high proportions of diatoms are 'unclassified' e.g. in JUR 544 diatoms out of 1149 (i.e. *c.* 50%) but also high at other sites (e.g. SHIC *c.* 25%), I assume these are the (L157) "unidentified (obscured, undiagnostic or indistinct)" diatoms. With so many in this category (I acknowledge when doing diatom analysis allocating some diatoms to this category is inevitable, but in my experience, these would be unusually high proportions) there needs to be a bit more detail as to why and whether this has any consequences. For example, if they are obscured then by what? What is meant by undiagnostic? And indistinct? Most diatoms can be identified to species level (or if not identified with confidence to a described species at least put into separate taxonomic units) so is this an issue with the technique, for example the use of SEM rather than light microscope? Some comments would be useful. What I am worried about here is some sort of systematic bias, for example if some genera are more undiagnostic than others due to being say harder to identify.

**Response:**

The high proportion of "obscured & undiagnostic" diatoms is due to the large amount of other insoluble particles & diatom fragments present in ice cores. The insoluble particulate matter greatly outnumber the diatoms (roughly 100:1 ratio) and frequently obscure part of the diatom remains (**See Figure 1 to Figure 6 below**). The large number of fragmented diatoms in snow and ice core samples has been widely documented, with fragments sometimes even exceeding the number of whole diatom frustules present in the sample (**Burckle et al., 1988; Budgeon et al., 2012; Delmonte et al., 2017; Allen et al., 2020; Tetzner et al., 2021**). This characteristic feature highlights how diatom records preserved in ice cores differ from the diatom records preserved in other archives (marine cores, lake sediments, etc). Since many diatom fragments do not retain diagnostic features, it is generally not possible to classify them (**See Figure 1 to Figure 6 below**). It is also important to note that there was no partiality in fragmentation nor concealment, based on the ratios of pennate to centric fragments and the varied morphologies of partially covered frustules. Therefore, it is unlikely that "obscured & undiagnostic" diatoms are causing a systematic bias.

We have added a detailed explanation of the above to better communicate the constraints encountered in analysing the diatoms from these AP/EL ice cores (Lines 177-185).

[Figure]

[Figure]

[Figure]

[Figure]

Figure 1 - 6. Micrographs showing examples of the insoluble particulate matter preserved in ice layers from the Jurassic ice core (JUR). The red rectangle highlights the presence of diatoms and diatom fragments partly obscured by inorganic insoluble particles. Scale bar in bottom right represents 50 microns.

**Prof V. Jones comment #7**

The percentage diatom assemblage data presented in Figures (2 -4) are "normalised" to the main species (i.e. recalculated to 100%) but when you have varying amounts of unidentified species it makes it difficult to compare between sites. Ideally, you need to look at the whole assemblage if not then there is an assumption that all the unidentified diatoms are a random subsample (not biased) of the assemblage? Is that a reasonable assumption?

I assume that as with light microscopy fragments are counted e.g. a fragment of a raphid diatom central area is usually distinct enough to assign to a genus if not a species. But if fragments form part of the 'indistinct' category then there is a need to square this with the pristine nature (ornamentation and chains are mentioned L409) of the diatoms. Again, this could be covered with more detail on what constitutes the 'unclassified' portion.

**Response:**

As previously mentioned (See response comment #6), the large number of "unclassified" species is due to the large number of obscured diatoms and undiagnostic diatom fragments, a characteristic feature in the diatom records preserved in ice cores. The recovery of specimens still articulated in short chains at the bottom of the ice cores evidence the diatoms we find in ice cores are not affected by mechanical fracturing after being deposited. This is also supported by lab experiments where diatom frustules have withstood pressures equivalent to the ones exerted at the bottom of an icesheet (**Hamm et al., 2003**). Therefore, supporting diatoms are not fragmented within the ice. Since the "undiagnostic" fragments are not likely to be produced in the ice, then, we are able to assume that they constitute a random subsample which does not bias the "normalisation" of the diatom assemblage. We acknowledge there will always be a potential bias when defining the main diatom assemblage. However, the assumptions we have taken do not contribute to increase the bias. Since the "unclassified" category contributes 18 to 52% of the total annual diatom abundance, we agree this must be acknowledged in the manuscript.

To address this comment we have modified the manuscript, emphasising the presence of fragments (Lines 460-468), including more details about the "unclassified" portion (Lines 175-185) and outlining some of the possible causes for diatom fragmentation before their deposition (Lines 463-468).

**Reference**

Hamm, C. E., Merkel, R., Springer, O., Jurkojc, P., Maier, C., Prechtel, K., & Smetacek, V. (2003). Architecture and material properties of diatom shells provide effective mechanical protection. Nature, 421(6925), 841-843.

**Prof V. Jones comment #8**

Fig 1 the locations of the ice cores aren't particularly clear – an inset would help here (like that of Figure 6..or just refer to that) and the caption needs to include the names of the ice cores. Locations such as Amundsen-Bellinghausen seas, Ellsworth Land need to be added.

**Response:**

We have modified Figure 1 and its caption as suggested. We acknowledge the locations of the ice cores in Figure 1 look close together. To address this comment we added a rectangle in Figure 1, directing the reader to Figure 6 to see the ice core locations from a closer perspective.
* * *
**Prof V. Jones comment #9**

What is the justification of the decadal subsets? if these are "to assess the consistency of the assemblage" (L160) there might be better ways to do that e.g. DCA axis scores to see if there are changes in 'turnover'. Subsequently there is more emphasis on using the decadal data sets to examine changes in diatom concentration (number per L) between the 2 time periods (L404) and these are compared directly but the time intervals aren't the same e.g. JUR is 1992-2001 and 2002-2012 whilst SHBL and SHIC are 1999-2008 and 2009-2019. So I am unclear as to the justification for these decadal means when all 3 cores show positive trends over the whole sampled period and section 2 doesn't discuss climate/sea ice changes at these time intervals.

**Response:**

Our reason to analyse temporal changes in the assemblages and on the diatom concentration was to assess the consistency of the diatom record in in response to recent environmental changes in the region. In particular, to determine if there were shorter-term shifts in the diatom concentrations caused by changes in certain diatom species or if there was a general shift in all diatom species that shape the main assemblage. We decided to assess this at a decadal timeframe to reduce the potential imprints of interannual variability. We did not discuss specific changes in climate/sea ice during the assessed decades because trends in both, wind strengthening and sea ice retreat, have been sustained over the last decades.

We acknowledge we did not specify that wind and sea ice trends have been sustained over time or refer to the different timing of the decadal subsets across the sites. To address this comment, we have modified the manuscript, specifying our reasons to analyse the dataset in decadal subsets (Lines 189-191), we have clarified that recent trends have been sustained over the last decades (Lines 471 and 479) and we specified the timeframes considered when comparing recent changes in the diatom concentration (Lines 454-455).
* * *
**Prof V. Jones comment #10**

Line 267 S. gracilis (not gracilics)

**Response:**

Revised as suggested (Line 298)